# Kinetochore- and chromosome-driven transition of microtubules into bundles promotes spindle assembly

Jurica Matković[1], Subhadip Ghosh[2], Mateja Ćosić[1], Susana Eibes[3], Marin Barišić [3,4], Nenad Pavin [2] & Iva M. Tolić [1] ✉

Mitotic spindle assembly is crucial for chromosome segregation and relies on bundles of microtubules that extend from the poles and overlap in the middle. However, how these structures form remains poorly understood. Here we show that overlap bundles arise through a network-to-bundles transition driven by kinetochores and chromosomes. STED super-resolution microscopy reveals that PRC1-crosslinked microtubules initially form loose arrays, which become rearranged into bundles. Kinetochores promote microtubule bundling by lateral binding via CENP-E/kinesin-7 in an Aurora B-regulated manner. Steric interactions between the bundle-associated chromosomes at the spindle midplane drive bundle separation and spindle widening. In agreement with experiments, theoretical modeling suggests that bundles arise through competing attractive and repulsive mechanisms. Finally, perturbation of overlap bundles leads to inefficient correction of erroneous kinetochore-microtubule attachments. Thus, kinetochores and chromosomes drive coarsening of a uniform microtubule array into overlap bundles, which promote not only spindle formation but also chromosome segregation fidelity.

Segregation of chromosomes during mitosis relies on the mitotic spindle, a supramolecular micro-machine made of microtubules and numerous microtubule-associated proteins, which is built de novo in each cell cycle[1]. While some spindle microtubules exist as individual filaments, such as astral microtubules, many are organized in bundles, such as kinetochore fibers and overlap bundles[2,3]. Kinetochore fibers are bundles of parallel microtubules that attach in an end-on manner to the kinetochore on the centromere of each chromosome, and link the kinetochore with the spindle pole[4,5]. Overlap or interpolar bundles consist of antiparallel microtubules that extend from the opposite spindle halves[6]. Spindle assembly has been extensively studied with focus on kinetochore dynamics and the formation of kinetochore fibers[7–12], whereas the mechanism of overlap bundle formation remains poorly understood.

Overlap bundles are essential for spindle assembly, as inhibition of the Eg5/kinesin-5 motor protein, which slides the antiparallel microtubules apart, leads to monopolar spindles[13–16]. Such spindles contain kinetochore fibers but not overlap bundles and cannot segregate the chromosomes, highlighting the indispensable role of overlap bundles in spindle functioning. In a mature spindle in metaphase, overlap bundles link sister kinetochore fibers like a bridge and because of this interaction are known as bridging fibers[17]. Their functions include balancing the tension on kinetochores[17–21], promoting chromosome alignment at the spindle midplane[22,23] and driving spindle elongation and chromosome segregation in anaphase[24–28]. Microtubules within bridging fibers are linked by the protein regulator of cytokinesis 1 (PRC1)[17,21,29], a crosslinker that has a 10-fold preference for antiparallel versus parallel microtubule overlaps in vitro[30–33]. Such crosslinkers together with motor proteins may drive the formation of

---

[1]Division of Molecular Biology, Ruđer Bošković Institute, Zagreb, Croatia. [2]Department of Physics, Faculty of Science, University of Zagreb, Zagreb, Croatia. [3]Cell Division and Cytoskeleton, Danish Cancer Society Research Center, Copenhagen, Denmark. [4]Department of Cellular and Molecular Medicine, Faculty of Health Sciences, University of Copenhagen, Copenhagen, Denmark. ✉e-mail: tolic@irb.hr

initial overlaps between two antiparallel microtubules, as shown by computer simulations[34–37].

Mitotic spindles of human cells are made of more than 6000 microtubules[38]. In general, microtubules present in large numbers can exist in different phases, from loose networks to tight bundles, where the networks can be isotropic if the filaments are unoriented, nematic if the filaments are parallel and point either way, or polar if they point in a common direction[39]. Different types of microtubule networks have been studied in vitro[40–42], together with theoretical studies that explored under what conditions certain types of polymer networks appear or undergo a transition to bundles[43–47]. Yet, whether similar transitions occur in cells and what biological function they may have are open questions.

In this work we develop an assay for bundle formation, together with a theoretical model, and show that microtubules crosslinked by PRC1 undergo a network-to-bundles transition during early stages of spindle assembly. We find that this transition is stimulated by kinetochores, which promote microtubule bundling by binding to them laterally via CENP-E/kinesin-7 in a manner regulated by Aurora B. We further show that the separation of the bundles, which leads to spindle widening and the characteristic spindle shape, is driven by steric interactions of the chromosomes bound to the bundles as they congress to the spindle midplane, based on our findings that spindles with uncondensed or incompletely congressed chromosomes are narrower. Moreover, we identify a function of overlap bundles in the correction of erroneous kinetochore-microtubule attachments. Thus, our experiments, supported by the theoretical model, reveal that kinetochores and chromosomes together with crosslinkers drive coarsening of an initially uniform microtubule array into neatly organized overlap bundles, which not only help spindle assembly but also promote error-free mitosis.

## Results

### Microtubules undergo a network-to-bundles transition

The overarching question of how the cell generates microtubule bundles that form the spindle shape contains two parts: how individual bundles are formed and what determines their separation. To study individual bundle formation, we first explored microtubule organization during early stages of mitosis by using stimulated emission depletion (STED) microscopy[48] to obtain super-resolution images of microtubules, together with the microtubule-crosslinker PRC1 and chromosomes (Fig. 1a). The spatial pattern of microtubules and PRC1 changed profoundly during spindle formation and maturation (Fig. 1a). In early prometaphase characterized by the "prometaphase rosette" where the chromosomes are arranged like flower petals along the edge of the nascent spindle[49,50], the majority of spindle body microtubules appeared as a diffuse array with a few bundles present mainly at the edges ("Early prometaphase" in Fig. 1a and Supplementary Fig. 1a). During late prometaphase, when chromosomes increasingly congress to the equatorial plane, and afterwards in metaphase, additional microtubule bundles appeared and their spacing became more regular, which was accompanied by the appearance of PRC1 stripes along the bundles ("Late prometaphase" and "Metaphase" in Fig. 1a and Supplementary Fig. 1a).

By looking at cross-sections of vertically oriented spindles, we found that microtubules of the spindle body are initially uniformly distributed over a ring-shaped region, and astral microtubules extend radially outwards (Fig. 1a end-on view). The spindle body microtubules became rearranged into discrete bundles that fill the spindle cross-section at an even spacing (end-on view in Fig. 1a and Supplementary Fig. 1a). Similarly, PRC1 distribution changed from a nearly homogeneous scattering over the ring-shaped region, which we call PRC1 network, to spot-like structures that colocalized with the microtubule bundles (end-on view in Fig. 1a and Supplementary Fig. 1a).

To study how a loose microtubule array transforms into bundles, we developed a live-cell assay termed "bundling assay" based on PRC1 as a marker of microtubule overlaps, where we imaged cross-sections of vertically oriented spindles in HeLa cells expressing PRC1-GFP starting at the rosette stage for 9 min at 5.4 s intervals (Fig. 1b, Supplementary Fig. 1b, c). The rosette was identified by a chromosome circle surrounding the spindle, which was not yet fully closed (Supplementary Fig. 1b top and Fig. 1a bottom left). Time-lapse images from the bundling assay revealed that the spatial distribution of PRC1 in the spindle cross-section gradually changes from a homogeneous dispersion over a ring-shaped area (PRC1 network) to discrete clusters (Fig. 1b and Supplementary Movie 1). Segmentation analysis (Methods; Supplementary Fig. 1d, e) showed that the number of PRC1-labeled segments and their mean PRC1-GFP intensity increased over time, whereas the segment area decreased (Fig. 1c, d and Supplementary Fig. 1f). The intense imaging protocol did not affect spindle functioning, given that the spindles subjected to the bundling assay entered anaphase at similar times to control spindles that were imaged at 5-min intervals (Supplementary Fig. 1g, h), and the network-to-bundles transition found in the bundling assay was consistent with STED images of PRC1-GFP (Fig. 1a and Supplementary Fig. 1a) and confocal images of untransfected cells immunostained for PRC1 (Supplementary Fig. 1i, j). The latter result also implies that the dynamics of PRC1-GFP reflects the dynamics of endogenous PRC1. Thus, a network-to-bundles transition of PRC1 occurs during fully functional spindle assembly.

Intensity profiles of PRC1 along the ring-shaped area indicate that in comparison with the nearly constant PRC1 intensity in the rosette, the intensity at a later stage exhibited higher peaks and deeper valleys ($t = 0$ vs. 7.2 min, Fig. 1e, f and Supplementary Fig. 1k). The mean intensity did not change substantially, but the standard deviation increased 2.6-fold ($n = 10$, Fig. 1g). This suggests that the nearly uniform PRC1-labeled microtubule network undergoes a transition to bundles that accumulate PRC1, leaving the space between the bundles almost free from PRC1.

We next asked how the initial PRC1 network is formed. A large fraction of spindle microtubules is nucleated by the augmin complex at the wall of other microtubules[51,52]. When we used the bundling assay on cells depleted of the augmin subunit Haus6, we found that, in contrast to untreated cells, a PRC1 network was absent at the rosette stage (Supplementary Fig. 2a–c and Supplementary Movie 2) and the dynamics of bundle formation was slower (Supplementary Fig. 2d–f). These results suggest that the initial array of microtubules crosslinked by PRC1 is largely nucleated by the augmin complex.

To verify that the redistribution of PRC1 indeed reflects changes of the microtubule network, which was observed in STED images, we first used nocodazole to depolymerize microtubules at the rosette stage. This treatment caused the disappearance of the PRC1 network (Supplementary Fig. 2g), indicating that PRC1 forming this structure is bound to microtubules. Second, we noticed that PRC1 colocalized with microtubules in the central part of the spindle but was excluded from astral microtubules throughout prometaphase (Fig. 1a and Supplementary Fig. 2h), suggesting that PRC1 selectively localizes to antiparallel microtubule overlaps already in early mitosis[53]. Third, we imaged spindles stained with SiR-tubulin and found that, similarly to PRC1, tubulin signal undergoes a change from a nearly uniform to a spotted distribution (Supplementary Fig. 2i, j). Yet, PRC1 is a better marker of antiparallel overlaps not only because of its specificity but also because of the lower background signal (Fig. 1b vs. Supplementary Fig. 2i). Moreover, an increase in the number of bundles during prometaphase was observed in horizontally oriented HeLa spindles labeled with PRC1-GFP, as well as in spindles of human non-transformed retinal pigment epithelial hTERT-RPE1 cells stained with SiR-tubulin (Supplementary Fig. 2k), suggesting that the process of

bundle formation is independent of the cell line and labeling. Taken together, these results confirm that the PRC1 redistribution reflects the redistribution of overlap microtubules from a nearly uniform network in early prometaphase to the regularly spaced overlap bundles in metaphase.

### Kinetochores promote overlap bundle formation via CENP-E

Overlap microtubule bundles crosslinked by PRC1 are found adjacent to kinetochores during the formation of kinetochore fibers in prometaphase[12], as well as in metaphase[54]. These findings inspired us to ask whether kinetochores play a role in overlap bundle formation during early prometaphase (Fig. 2a), when kinetochores are predominantly laterally attached to spindle microtubules[49,50,55,56]. We found that, in a rosette cross-section of unperturbed spindles, most of the PRC1-labeled bundles had an adjacent kinetochore at its outer side (Fig. 2b). The bundles with

an associated kinetochore had a higher PRC1 signal intensity than those without a kinetochore (Fig. 2c), consistent with the hypothesis that kinetochores promote microtubule bundling.

To explore the role of lateral attachments of kinetochores in bundle formation, we inhibited Aurora B, which is in prometaphase localized mainly at centromeres (Supplementary Fig. 3a) and required for the lateral attachment[50,57–59]. We first explored the effect of Aurora B inhibition by barasertib (AZD1152-HQPA)[60,61] on mature bundles in metaphase. Strikingly, we found a 4.2-fold decrease in the signal intensity of individual PRC1-labeled bundles in the cross-section of metaphase spindles (n = 452 bundles from 14 cells, Fig. 2d, e; Supplementary Fig. 3b and side view in Supplementary Fig. 3c), suggesting that antiparallel bundles contained fewer microtubules. To confirm this, we measured the tubulin signal intensity in the central part of individual bridging fibers in the region between sister kinetochores in metaphase and found a 56.3%

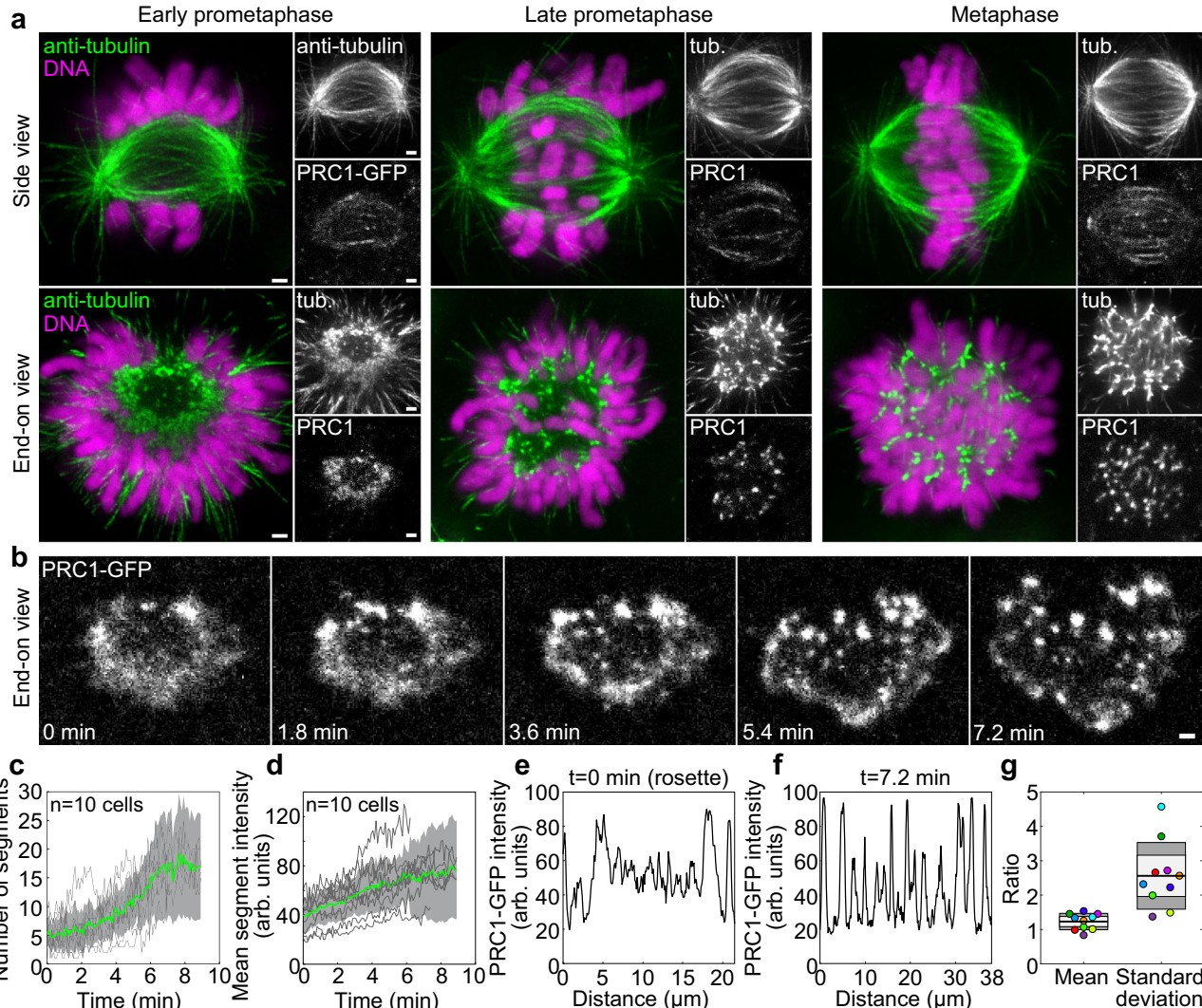

**Fig. 1 | PRC1-crosslinked microtubules undergo a transition from a loose array to bundles during spindle assembly. a** Spindles immunostained for α-tubulin and imaged at STED resolution in a HeLa-Kyoto BAC cell line stably expressing PRC1-GFP with DNA stained by DAPI (both imaged at a confocal resolution), in early prometaphase, late prometaphase, and metaphase. Spindles lying parallel (first row) or perpendicular (second row) to the imaging plane are shown. **b** Time-lapse images (single plane) of a cross-section of a vertically oriented prometaphase spindle in a HeLa-Kyoto BAC cell stably expressing PRC1-GFP, starting with the prometaphase rosette. **c** Number of PRC1 segments in the midplane over time for cells as in (**b**), obtained using Squassh segmentation. Gray lines represent

individual cells, green line the mean and gray areas the standard deviation, n = 10 cells in 10 independent experiments. **d** Mean intensity of PRC1 segments from the same cells as in (**c**). **e, f** Intensity line plots of PRC1-GFP in the cell from (**b**) at t = 0 (**e**) and t = 7.2 min (**f**) along the periphery of the spindle cross-section (lines shown in Supplementary Fig. 1k). **g** Ratio of mean PRC1-GFP intensity in (**f, e**) ("Mean") and ratio of the standard deviation in (**f, e**) ("Standard deviation"). The black line shows the mean; the light and dark gray areas mark 95% confidence interval on the mean and standard deviation, respectively; n = 10 cells in 10 independent experiments. Dots of the same color represent the same cells. All scale bars, 1 μm.

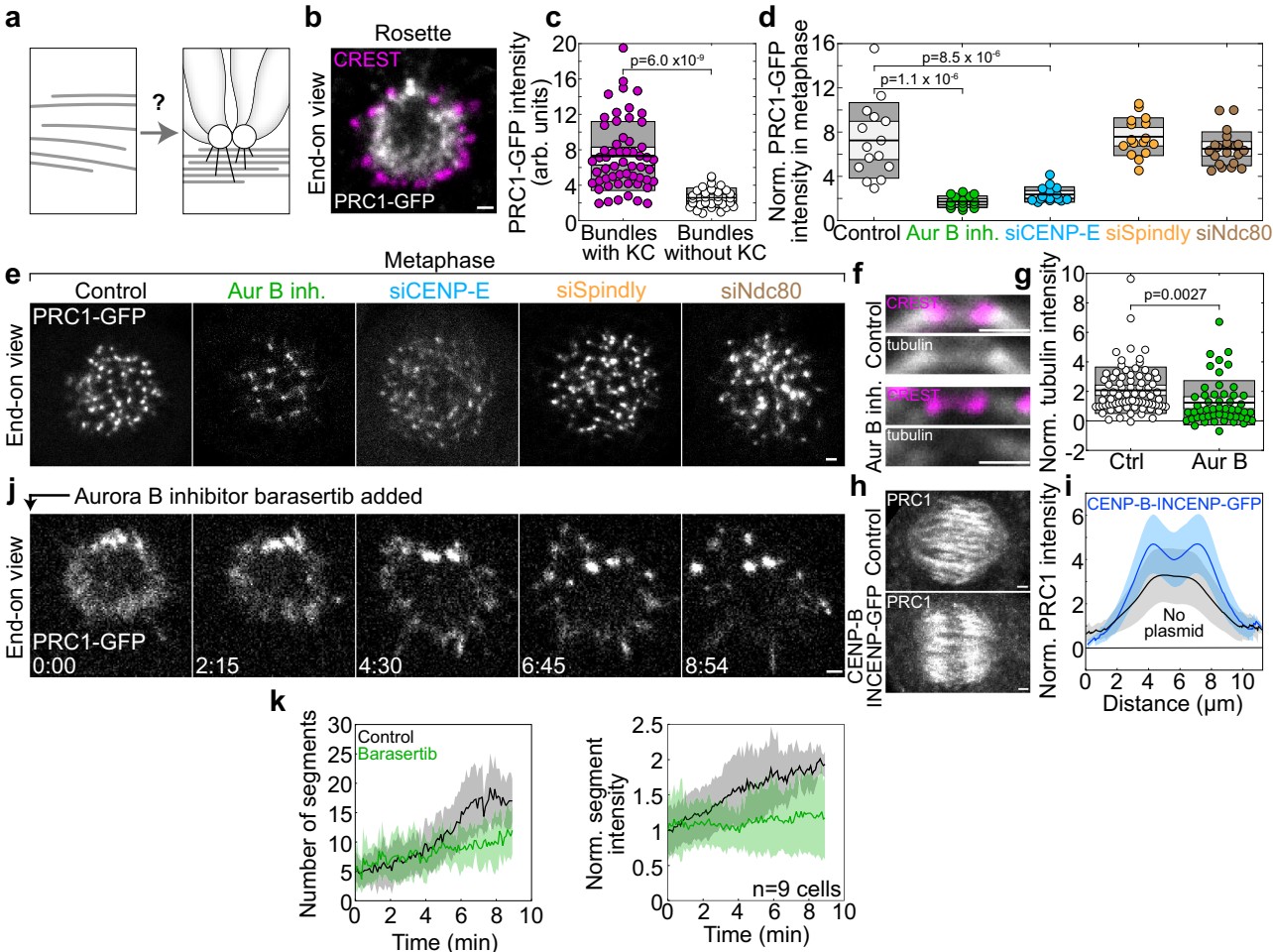

**Fig. 2 | Kinetochores promote microtubule bundle formation in an Aurora B-dependent manner. a** Scheme of kinetochore-driven bundle formation. **b** Midplane of a vertical spindle immunostained for CREST (magenta) in a HeLa-Kyoto cell expressing PRC1-GFP (white). **c** PRC1-GFP intensity of bundles with and without adjacent kinetochore, $n = 57$ and 30 bundles, respectively. **d** PRC1-GFP intensity of bundles in vertical metaphase spindles (circles represent the mean per spindle), normalized to the cytoplasmic PRC1-GFP intensity. Numbers of bundles/cells from left to right: 827/15, 452/14, 730/15, 880/16, 774/21; $p$ values are given for the significant differences from control, one-way ANOVA test. **e** Vertical metaphase spindles in HeLa-Kyoto cells expressing PRC1-GFP (white), treated as indicated. **f** Region around a kinetochore pair immunostained for CREST (magenta) and tubulin (white) in an untransfected HeLa cell, control (top) and after Aurora B inhibition (bottom), merged channels and only tubulin. **g** Mean tubulin intensity at the bridging fiber center in control and Aurora B-inhibited cells, normalized to the cytoplasmatic anti-tubulin intensity; $n = 78/10$ bundles/cells (control) and 52/9 (inhibition). **h** Maximum-intensity projections of metaphase spindles immunostained for PRC1 (white) in an untransfected HeLa cell (control, top), and a cell transfected with CENP-B-INCENP-GFP (bottom). **i** Line intensity profiles of the anti-PRC1 signal intensity, normalized to the cytoplasmatic intensity, along the spindle axis, for control ($n = 33$) and treated cells as in (**h**) ($n = 30$), mean (central line) and standard deviation (colored surfaces). The higher anti-PRC1 signal after transfection with CENP-B-INCENP-GFP is likely due to INCENP overexpression. **j** Midplane of a vertical prometaphase spindle in a HeLa-Kyoto cell expressing PRC1-GFP (white), after addition of Barasertib. **k** Number of PRC1 segments (left) and normalized (to the intensity at $t = 0$) mean intensity of PRC1 segments (right) from squassh segmentation in control and Barasertib-treated cells. Colored surfaces around the central lines (mean) represent standard deviation, $n = 9$ cells. In (**c**, **d**, **g**), the black line shows the mean; the light and dark gray areas mark 95% confidence interval on the mean and standard deviation, respectively. In (**c**, **g**), $p$ value from a two-tailed t-test is given. In all panels, at least 3 independent experiments per condition; scale bars, 1 μm.

decrease upon Aurora B inhibition, on average (Fig. 2f, g and Supplementary Fig. 3d). In 13.5% (7 out of 52) of these regions between sister kinetochores a bridging fiber was undetectable, i.e., its signal was below the background, and this fraction was larger than in untreated cells, where bridging fibers were undetectable in 1.3% cases (1 out of 78, Fig. 2g). These results suggest that Aurora B activity regulates the number of microtubules within the PRC1-labeled bridging fibers in metaphase.

We addressed the role of Aurora B from another angle by exploring localization relationships between Aurora B and PRC1 in metaphase. We found that the signal intensity of PRC1 within the bridging fiber is correlated with the signal of Aurora B in the neighboring centromere (Supplementary Fig. 3e). Moreover, when we displaced Aurora B from the inner centromere to a position closer to the kinetochore by transfecting cells with CENP-B-INCENP-GFP[62–64], we observed a remarkable change in the PRC1 signal distribution. Instead of a single broad PRC1 peak in the central part of the spindle that is evident in untreated cells, the cells with displaced Aurora B showed two PRC1 peaks about 1 μm away from the spindle equator on either side (Fig. 2h, i and Supplementary Fig. 3f). These results suggest that Aurora B regulates the localization of PRC1-labeled bundles within the spindle.

To test the effect of Aurora B on the dynamics of bundle formation during early prometaphase, we combined the bundling assay with acute inhibition of Aurora B by barasertib added at the rosette stage (Fig. 2j and Supplementary Movie 3). Aurora B inhibition led to a slower increase in the number and intensity of PRC1-labeled bundles over time in comparison with untreated cells (Fig. 2k and Supplementary

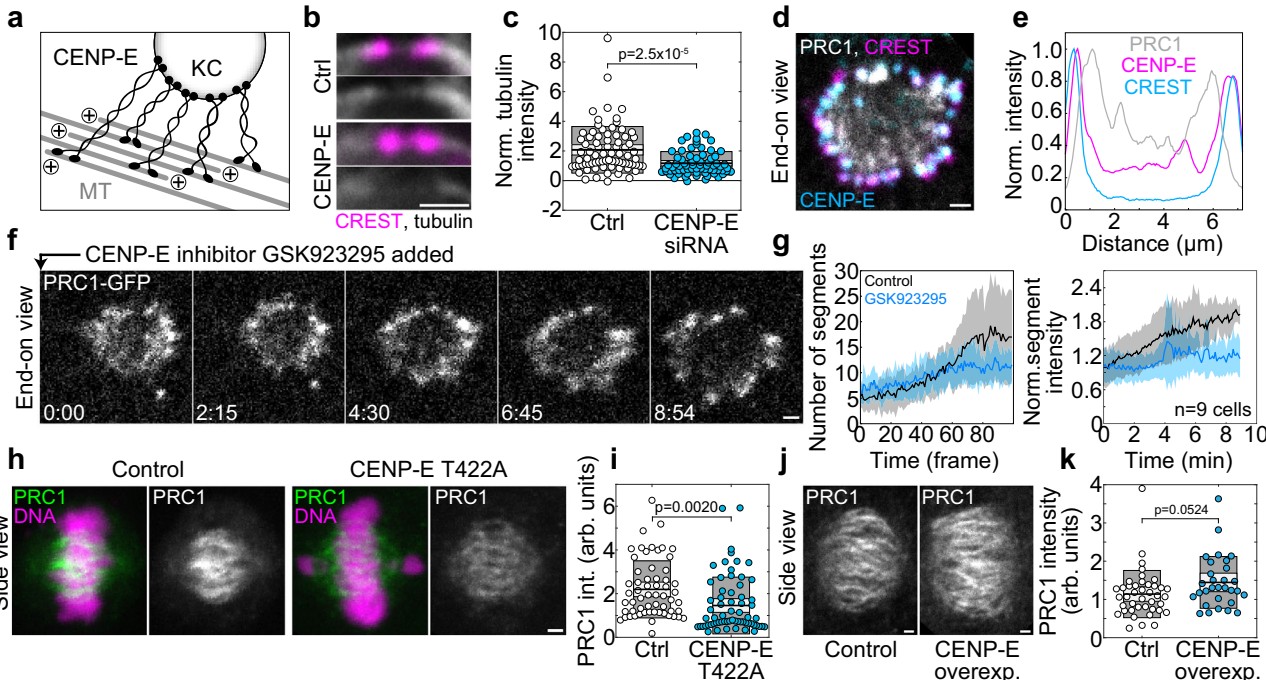

**Fig. 3 | CENP-E promotes overlap bundle formation. a** Scheme of CENP-E-driven microtubule bundling. **b** Region around a kinetochore pair immunostained for CREST (magenta) and tubulin (white) in an untransfected HeLa cell, control (top) and after CENP-E depletion (bottom), merged channels and only tubulin. **c** Mean tubulin intensity at the bridging fiber center in control and CENP-E depleted cells, normalized to the cytoplasmatic anti-tubulin intensity; $n = 78/10$ bundles/cells (control) and 70/14 (CENP-E depletion), from 3 independent immunostainings. **d** Cross-section of a vertical prometaphase spindle immunostained for CREST (magenta) and CENP-E (cyan) in a HeLa-Kyoto cell expressing PRC1-GFP (white). **e** Line intensity plots of PRC1-GFP, CREST and CENP-E from the spindle shown in d, normalized to the corresponding maximum value. The line was drawn through the spindle cross-section center, with a thickness of 1/3 of the spindle cross-section. **f** Time-lapse images of a midplane of a vertical prometaphase spindle in a HeLa-Kyoto cell expressing PRC1-GFP (white), after addition of the CENP-E inhibitor GSK-923295. **g** Number of PRC1 segments (left) and normalized mean intensity of PRC1 segments (normalized to the intensity at $t = 0$, right) over time from squassh segmentation in control and GSK-923295-treated cells, as indicated. Colored surfaces around the central lines (mean) represent standard deviation, $n = 9$ cells from 9 independent experiments. **h** Horizontally oriented metaphase spindles immunostained for PRC1 (white) in a U2OS doxycycline-induced CENP-E wild-type (control, left) and T422A CENP-E mutant cell in which endogenous CENP-E was silenced (right). **i** Total PRC1 intensity on the spindle in cells expressing wild-type CENP-E (control, $n = 59$) and T422A CENP-E mutant ($n = 66$) from (**h**), from 5 immunostainings. **j** Horizontally oriented metaphase spindles immunostained for PRC1 (white) in an untransfected HeLa cell line (control, left) and after CENP-E overexpression (right). **k** Total PRC1 intensity on the spindle in control ($n = 41$) and CENP-E overexpression ($n = 30$), from 3 immunostainings. In (**c, i, k**), the black line shows the mean; the light and dark gray areas mark 95% confidence interval on the mean and standard deviation, respectively, and $p$ values from a two-tailed t-test are given. All scale bars, 1 μm.

Fig. 3g), which confirmed that Aurora B regulates the dynamics of bundle formation.

The mechanism by which Aurora B at the kinetochore promotes microtubule bundling likely relies on Aurora B substrates involved in microtubule binding. We hypothesize that if microtubule-binding proteins that are localized at the kinetochore in multiple copies attach to several microtubules positioned close to the kinetochore, this could promote microtubule bundling (Fig. 3a). Candidates for this activity are two motor proteins, CENP-E (kinesin-7) and cytoplasmic dynein, and the outer kinetochore protein Ndc80, given that they bind to microtubules, localize at the kinetochore in prometaphase (CENP-E and dynein) or throughout mitosis (Ndc80), and are regulated by Aurora B[50,65–67].

We first analyzed the signal intensity of PRC1-labeled bundles in metaphase after depletion of these candidates. CENP-E depletion by siRNA resulted in a 3.1-fold lower intensity of PRC1 in individual bundles in metaphase ($n = 730$ bundles from 14 cells, Fig. 2d, e and Supplementary Fig. 3b, c; see Supplementary Fig. 4 for depletion efficiency) and a 59.3% decrease in the tubulin signal intensity in the central part of individual bridging fibers (Fig. 3b, c and Supplementary Fig. 3d), where in 1.4% cases bridging fibers were undetectable (1 out of 70, Fig. 3c). These results indicate that CENP-E is required for proper bundle formation and motivated us to explore its localization and dynamics during early prometaphase. In the rosette, we found that CENP-E localizes at the side of the kinetochore facing the PRC1-labeled

bundles (Fig. 3d, e), which suggests that it links the kinetochore with the antiparallel bundles[53]. The bundling assay revealed that acute inhibition of CENP-E by GSK-923295[68] at the rosette stage led to a slower increase in the number and intensity of PRC1-labeled bundles over time than in untreated cells (Fig. 3f, g, Supplementary Fig. 3h and Supplementary Movie 4), supporting a role of CENP-E in bundle formation.

Our hypothesis that CENP-E activity, regulated by Aurora B phosphorylation, promotes microtubule bundling predicts that the non-phosphorylatable T422A CENP-E mutant[65] should lead to decreased bundling. Indeed, replacing endogenous CENP-E with the T422A mutant resulted in a 34% smaller amount of PRC1 on the metaphase spindle, indicating reduced microtubule bundling ($n = 66$ cells from 5 independent experiments, Fig. 3h, i; see Supplementary Fig. 4 for depletion of endogenous CENP-E). Furthermore, the hypothesis that CENP-E promotes bundling predicts that over-expression of CENP-E should lead to increased bundling, which we indeed observed as a higher intensity of PRC1 on the metaphase spindle upon CENP-E overexpression (Fig. 3j, k and Supplementary Fig. 3i). Taken together, these results indicate that kinetochores promote bundle formation by lateral binding to microtubules via CENP-E in an Aurora B-regulated manner.

To test whether dynein at the kinetochore is required for overlap bundle formation, we depleted Spindly, a kinetochore-specific adaptor for dynein that recruits dynein to the kinetochore[69]. Spindly depletion

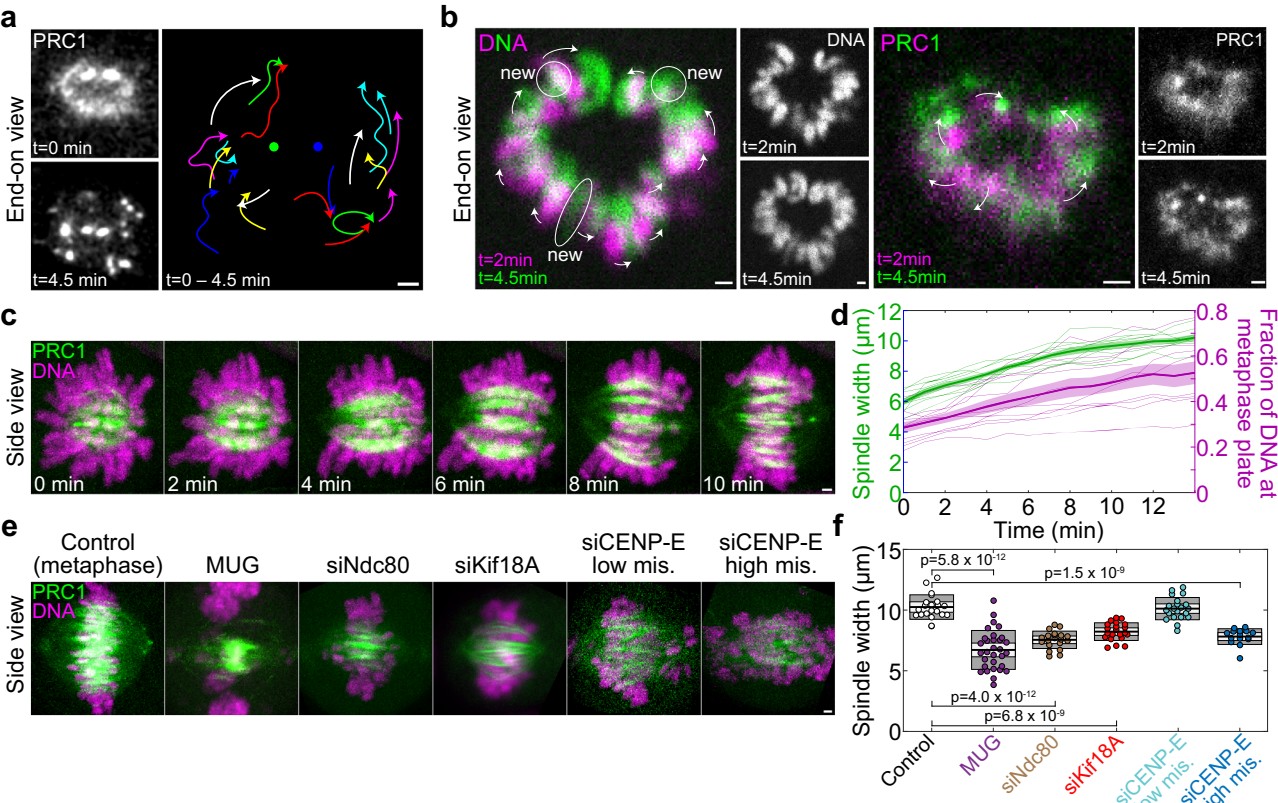

**Fig. 4 | Chromosomes separate microtubule bundles and increase spindle width.** **a** Left, Vertical prometaphase spindle midplane in a HeLa-Kyoto cell expressing PRC1-GFP (white), before ($t = 0$) and after antiparallel bundle formation ($t = 4.5$ min). Right, trajectories of all bundles from time 0 until 4.5 min (arrowheads), where the points between the end points were smoothed for better visibility and raw data are shown in Supplementary Movie 5. Two large bundles visible at the top at time 0 are fixed in space (central points). Time interval between images is 2.7 s. Similar results were obtained in 8 independent experiments. **b** Midplane of a vertical prometaphase spindle, showing DNA stained with SiR-DNA (left block) or PRC1-GFP (right block), as indicated, in a HeLa-Kyoto cell expressing PRC1-GFP, at $t = 2$ min (magenta) and $t = 4.5$ min (green). Individual images are shown in white at the right of the merged images. Encircled areas marked "new" denote the chromosomes that arrived to the spindle midplane between $t = 2$ min and $t = 4.5$ min, arrows denote the movement of DNA (left) or PRC1 bundles (right). Similar results were obtained in 5 independent experiments. **c** Time-lapse images (sum of 41 planes) of a horizontally oriented spindle in a HeLa-Kyoto cell expressing PRC1-GFP (green), and DNA stained with SiR-DNA (magenta) starting at early prometaphase. **d** Spindle width and fraction of DNA at the metaphase plate over time. Colored surfaces around the central lines (mean) represent standard deviation, $n = 10$ cells (thin lines) from 10 independent experiments. **e** Live-cell images (maximum-intensity projections of 41 planes) of metaphase spindles in HeLa-Kyoto cells expressing PRC1-GFP (green), and DNA stained with SiR-DNA (magenta), treated as indicated. **f** Spindle width for the indicated treatments. The black line shows the mean; the light and dark gray areas mark 95% confidence interval on the mean and standard deviation, respectively; $p$ values for significant differences from control are shown, one-way ANOVA test. Number of cells was 15–31 per group. Data for Kif18A siRNA are replotted from ref. 77. All scale bars, 1 μm.

did not change the signal intensity of PRC1-labeled bundles in metaphase (Fig. 2d, e and Supplementary Fig. 3b; see Supplementary Fig. 4 for depletion efficiency), suggesting that dynein is not crucial for overlap bundle formation. Similarly, Ndc80 depletion did not alter bundle intensity (Fig. 2d, e and Supplementary Fig. 2b; see Supplementary Fig. 4 for depletion efficiency), arguing against a major role of Ndc80 in overlap bundle formation. Moreover, as Ndc80 is required for the establishment of end-on attachments of kinetochores to microtubules and thus the formation of kinetochore fibers[57,66,70,71], these data imply that overlap microtubule bundles can form independently of kinetochore fibers[53].

### Steric interactions of the chromosomes separate the bundles

As the bundles form, they undergo characteristic movements. We first focused on the bundles that connect the centrosomes in a straight line, which can be identified by the highest PRC1 signal and a lack of an attached chromosome at the rosette stage (see Fig. 1a). These bundles are typically found at the edge of the rosette and become centrally positioned over a period of about 5 min (Fig. 4a, images to the left). We traced the movement of all bundles and found that, in a coordinate system centered at the

brightest bundles, the bundles forming adjacent to them move outwards and encircle them, whereas the bundles forming on the opposite side of the cross-section move away from the brightest bundles (Fig. 4a, traces and Supplementary Movie 5). As this movement is accompanied by the arrival of more and more chromosomes to the spindle midplane, we hypothesized that chromosomes promote bundle separation and thus spindle width expansion by steric effects (crowding) upon congression to the midplane. Indeed, we found that as new chromosomes arrived at the midplane, the chromosomes that were already there moved away from the site of the new arrival (Fig. 4b left). This led to the overall outward movement of the PRC1-labeled bundles and spindle widening (Fig. 4b right and Supplementary Movie 6). When looking at the spindles in a side view, we found that the spindle width increases as increasingly more chromosomes arrive to the metaphase plate (Fig. 4c, d and Supplementary Movie 7). These results support the hypothesis that chromosome crowding increases bundle separation.

This hypothesis predicts that removing the chromosomes from the spindle and/or decondensing the chromatin should result in narrower spindles. To test this idea, we induced mitosis with unreplicated

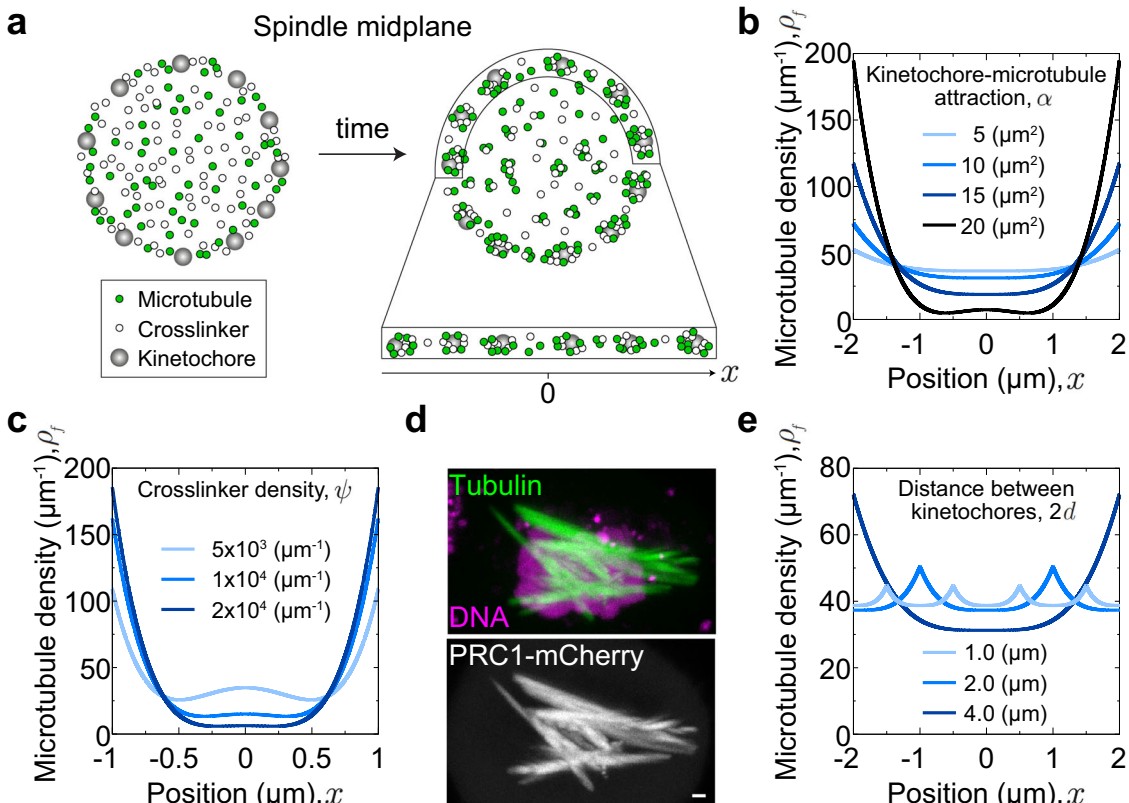

**Fig. 5 | Theoretical model of bundle formation. a** Schematic representation of the model. Kinetochores (gray) are equally spaced along the edge of the spindle midplane. Microtubules (green) and crosslinkers (white) are initially uniformly distributed and gather into bundles over time. **b** Steady-state microtubule density, $\rho_f$, for 4 values of the kinetochore-microtubule attraction, $\alpha$, as indicated. Other parameters are: $\psi = 10000\,\mu m^{-1}$, $d = 2\,\mu m$, $\kappa = 10\,\mu m^3$, $w = 1\,\mu m^3$. The kinetochores are positioned at $x = \pm d$. **c** Steady-state microtubule density, $\rho_f$, for 3 values of the crosslinker density, $\psi$, as indicated; $\alpha = 20\,\mu m^2$, $d = 1\,\mu m$, $\kappa = 5\,\mu m^3$, $w = 0.01\,\mu m^3$. **d** Spindle in a HeLa-TDS cell with overexpressed PRC1. Top: microtubules (SiR-tubulin, green) and DNA (Hoechst 33342, magenta), bottom: PRC1-mCherry (white). Scale bar, $1\,\mu m$. Similar observations were made in more than 10 cells from 3 independent experiments. **e** Steady-state microtubule density, $\rho_f$, for 3 values of the distance between kinetochores, $2d$, as indicated; $\alpha = 10\,\mu m^2$, $\psi = 10000\,\mu m^{-1}$, $\kappa = 10\,\mu m^3$, $w = 1\,\mu m^3$. Other parameters are kept fixed at $\zeta = 0.016\,\mu m^2$, $\bar{\omega}_{on} = 0.8\,\mu m^{-1}s^{-1}$, $\omega_{off} = 0.02\,s^{-1}$, $M_\rho = 0.02\,\mu m \bullet s^{-1}$. Determination of the parameters is described in Methods.

genome (MUG) by treating the cells with hydroxyurea to inhibit DNA replication and caffeine to override the DNA damage checkpoint[10,72], because in such cells the bulk of the uncondensed chromatin remains away from the spindle, while the kinetochores are attached to the spindle microtubules[10]. We found that the MUG spindles were significantly narrower with more tightly packed bundles than spindles in untreated cells (Fig. 4e, f).

Another prediction of the chromosome crowding hypothesis is that if the chromosomes are not aligned at the equator but distributed over the spindle, the spindles should be narrower, which we tested by several protein depletions. Ndc80 depletion, which results in a large fraction of chromosomes distributed along the spindle due to the absence of kinetochore fibers[70,73,74], led to narrower spindles than in untreated cells (Fig. 4e, f). To test chromosome misalignment without abolishing kinetochore fibers, we used depletion of Kif18A (kinesin-8), a motor protein that regulates chromosome oscillations around the spindle midplane, whose depletion leads to extensive oscillations and hence chromosome scattering along the spindle[75,76]. In this case we also found narrower spindles[77] (Fig. 4e, f). To further test our hypothesis, we divided the cells depleted of CENP-E into two groups: those with high chromosome misalignment, where more than 50% of the total chromosome mass was found away from the spindle midplane, and low misalignment where 1–5 chromosomes were misaligned. CENP-E depleted spindles with high misalignment were narrower than those with low misalignment (Fig. 4e, f). Taken together,

these results support the hypothesis that chromosome accumulation on the metaphase plate promotes the separation of the associated microtubule bundles and spindle width expansion.

### Theoretical model of bundle formation

To understand the physics of bundle formation, we develop a theoretical model that allows us to identify the conditions that lead to the formation of multiple bundles, each bound to a kinetochore. We aim for a minimal model that includes key interactions between microtubules, kinetochores, chromosomes, and crosslinkers. As these interactions are complex, a minimal model is a useful tool to understand the interplay between these multiple components and their roles in bundle formation, and to strengthen the hypotheses by comparison with experiments.

In our one-dimensional approach, representing the rim of the spindle midplane, we describe kinetochores as discrete points and explore the spatial distribution of microtubules (Fig. 5a). Kinetochores are considered as attractive points for microtubules, where multiple microtubules can bind and form a bundle. Kinetochores are equidistantly spaced, and we find stable equilibria for this choice of kinetochore distributions and reasonable parameters. Microtubules experience mutual local attraction due to crosslinkers and intermicrotubule repulsion due to excluded volume effects. Finally, steric interactions between chromosomes and microtubules are described as a nonlocal inter-microtubule repulsion. A description of the model is given in Methods.

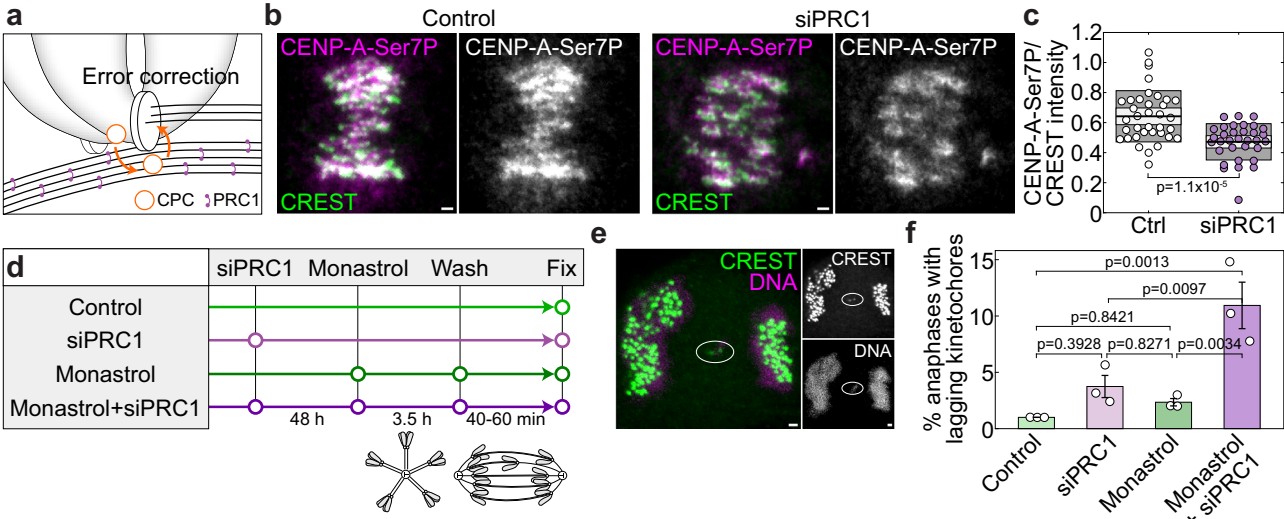

**Fig. 6 | PRC1-bound overlap bundles promote chromosome segregation fidelity. a** Scheme depicting the hypothesis that antiparallel overlaps help Aurora B to reach the kinetochore components, phosphorylate them and correct erroneous attachments. **b** Images showing anti-CENP-A-Ser7P (magenta) and CREST (green) in control (left) and PRC1-depleted HeLa cells (right), merged channels and only anti-CENP-A-Ser7P in white are shown. **c** Ratio between anti-CENP-A-Ser7P and CREST in control ($n = 36$) and PRC1-depleted cells ($n = 33$), from 3 immunostaining experiments. The black line shows the mean; the light and dark gray areas mark 95% confidence interval on the mean and standard deviation, respectively. **d** Scheme of the protocol in which monastrol washout is used to increases erroneous attachments in prometaphase. **e** Lagging chromosome (encircled) in anaphase in a PRC1-depleted HeLa cell after monastrol washout. Left, merged image of CREST (green) and DNA (magenta). Right, individual channels (white). **f** Frequency of lagging kinetochores in control (3 anaphases with lagging kinetochores/299 total anaphases), PRC1 siRNA (8/231), monastrol washout (7/298), and PRC1siRNA +monatrol washout (28/259) from 3 immunostaining experiments per condition. One-way ANOVA test showed significant difference between group means, *p* values are shown. Error bars represent s.e.m. The cell line used in this figure was unlabeled HeLa-TDS. All scale bars, 1 μm.

The solutions of the model show that kinetochores play an important role in the formation of microtubule bundles, with peaks of microtubule density at their positions (Fig. 5b). These peaks increase for an increasing kinetochore-microtubule attraction, which is accompanied by a decrease of the microtubule density between the kinetochores. A decrease in this attraction leads to an almost homogeneous microtubule distribution. A similar behavior was observed in experiments, where depletion of CENP-E led to thinner bundles and its overexpression to thicker bundles (Figs. 2d, e, 3j, k). Thus, theory together with experiments suggest that microtubules respond to the kinetochore-microtubule attraction by rearranging themselves between a uniform distribution and pronounced bundles at kinetochores.

Even though kinetochores are major attraction sites, microtubule crosslinkers also affect the formation of bundles (Fig. 5c). An increase in the crosslinker concentration leads to more pronounced bundles at the kinetochores, showing a similar trend as the increase in kinetochore-microtubule attraction. In agreement with this theoretical result, experiments in which PRC1 was overexpressed showed excessively thick overlap microtubule bundles (Fig. 5d and Supplementary Fig. 5). Thus, microtubule crosslinkers act cooperatively with kinetochores in the formation of bundles.

Finally, we explored how the distance between kinetochores influences microtubule distributions. When we increased the distance, the kinetochores attract more microtubules, whereas the central region between kinetochores contains fewer microtubules than for smaller inter-kinetochore distance (Fig. 5e). We assume that the cases with smaller and larger distances correspond to the transition of the spindle from prometaphase to metaphase, because during this transition increasingly more kinetochores leave the rim of the spindle midplane and thus their nearest neighbor distance along the rim increases. Indeed, we observed in experiments that in early prometaphase the microtubule distribution was roughly uniform, whereas at a later phase higher peaks and deeper valleys appeared (Fig. 1e, f). Taken

together, these results support our main hypothesis that the attractive interactions due to kinetochores and crosslinkers drive the formation of microtubule bundles.

## PRC1-labeled bundles promote chromosome segregation fidelity

Finally, we addressed the role of overlap bundles in chromosome segregation fidelity. It has been shown that in prometaphase microtubules adjacent to the centromeres stimulate kinetochore phosphorylation and thus correction of erroneous kinetochore-microtubule attachments through microtubule binding of Borealin, a component of the Chromosome Passenger Complex[71]. We hypothesized that PRC1-bound overlap bundles are required for error correction, in a manner that Aurora B not only promotes the formation of these bundles (Fig. 2), but also uses them as tracks for the movement towards kinetochore targets to correct the wrong attachments (Fig. 6a). This hypothesis predicts that reduction of overlap microtubules should lead to a decrease in kinetochore phosphorylation and an increase in chromosome segregation errors.

To test these predictions, we reduced the overlap bundles by depleting PRC1, as this is the most specific way to reduce these bundles known so far, which reduces overlap bundle microtubules by about 50% without affecting kinetochore fibers[22]. We checked the phosphorylation of serine 7 (Ser7) of CENP-A, a known Aurora B substrate[78], and found a significant drop in phosphorylation after PRC1 depletion (Fig. 6b, c), which suggests that PRC1-crosslinked overlap microtubules promote kinetochore phosphorylation.

To assess the role of overlap bundles in chromosome segregation fidelity, we first analyzed lagging kinetochores in anaphase and found that PRC1 depletion caused a small increase in lagging kinetochores over control cells, from 1% to 3.5% (Fig. 6d, f; see Supplementary Fig. 4 for depletion efficiency)[22]. To directly examine the function of overlap bundles in the correction of incorrect kinetochore attachments, we treated the cells with monastrol to block the spindles in a monopolar

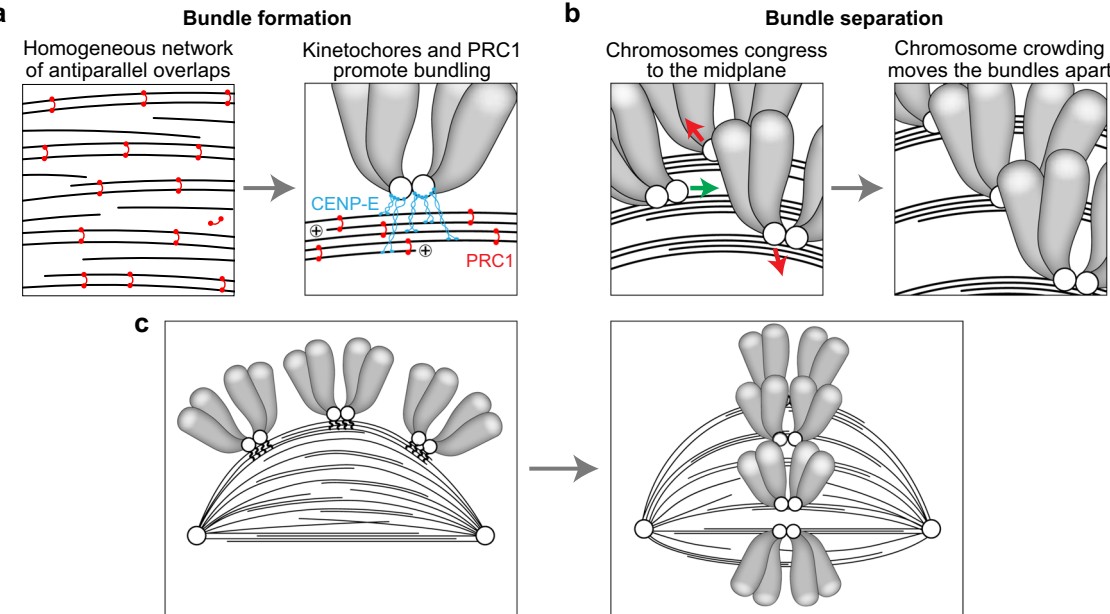

**Fig. 7 | Model for kinetochore-driven overlap formation. a** A loose microtubule network consisting of uniformly distributed antiparallel overlaps linked by PRC1 (red) undergoes a transition to tight bundles due to the microtubule-binding activity of kinetochores (CENP-E, blue). **b** Bundles are separated (red arrows) during chromosome congression (green arrow) to the spindle midplane and steric effects of neighboring chromosome arms. **c** The processes depicted in a and b lead to a transition from the prometaphase rosette (left), with a loose network of microtubules and laterally attached chromosomes, to a more mature spindle in late prometaphase (right), with distinct and separated bundles.

state with high incidence of syntelic attachments, where both sister kinetochores are attached to the same spindle pole[15], and washed out monastrol to allow for spindle assembly (Fig. 6d). Immunofluorescence analysis of spindles in anaphase showed that only 2.3% of monastrol-treated cells that were not depleted of PRC1 had lagging kinetochores (Fig. 6f), indicating that error correction operates in cells with proper overlap bundles. In contrast, monastrol-treated cells that were depleted of PRC1 had a significantly higher incidence of lagging kinetochores, 10.8% (Fig. 6e, f), revealing a role of overlap bundles in error correction. Taken together, these results suggest that PRC1-bound overlap bundles have a function in correcting erroneous kinetochore-microtubule attachments by promoting Aurora B-driven kinetochore phosphorylation.

## Discussion

By combining STED microscopy with a live-cell imaging protocol termed bundling assay, we have shown that spindle assembly occurs through a transition in which the initially uniform microtubule array is remodeled into bundles (Fig. 7a–c). Our experiments suggest that the temporal evolution of the bundles starts with the initial dilute network of antiparallel overlaps connecting the spindle poles, which relies on augmin-mediated microtubule nucleation together with the cross-linking activity of PRC1 and possibly other crosslinkers. When a kinetochore simultaneously attaches to the wall of several of these microtubules via CENP-E, this promotes further bundling of increasingly more microtubules by crosslinkers. While the chromosomes progressively congress to the spindle equator, they crowd there and thus move together with the attached bundles away from each other within the equatorial plane. As a result, the spindle cross-section widens and becomes uniformly filled with bundles and the attached chromosomes, and the spindle attains its metaphase shape.

We propose several competing mechanisms that are in action during transition to bundles. There are two attractive and two repulsive effects between the microtubules. Microtubules get attracted to neighboring microtubules via the crosslinkers (Fig. 7a). This process is facilitated locally by kinetochores because they attach to several

adjacent microtubules, which brings the microtubules close to one another and promotes their bundling by crosslinkers (Fig. 7a). Microtubules experience local repulsion from neighbors due to excluded volume effects, and a non-local repulsion due to chromosomes whose crowding at the spindle midplane moves the attached bundles away from each other (Fig. 7b). This non-local repulsion helps to prevent all bundles from ending up as a single thick bundle. A bundling phase transition has been studied in theoretical models of cytoskeleton networks[79,80], but here we show that additional interactions such as those between microtubules and kinetochores are important because they promote bundling. Yet, given that our model is one-dimensional, extensions of the model that would include inter-microtubule orientation may bring about new features of bundle formation.

Our experiments revealed that kinetochores actively promote the formation of antiparallel bundles in prometaphase. Kinetochores have been previously implicated in the formation of antiparallel bundles in different contexts, for example during meiosis I in mouse oocytes, where PRC1 is localized to the kinetochore through interaction with Ndc80, and facilitates bundling of antiparallel overlaps in vicinity of the kinetochore[81]. As we did not find PRC1 at the kinetochore in human somatic cells, the molecular mechanism of bundle formation may differ in these two systems, though the biophysical mechanisms may be similar. In anaphase in *C. elegans* zygotes, formation of antiparallel bundles of the central spindle requires the kinetochore proteins KNL-1, BUB-1, CENP-F, and CLASP[82]. CENP-E, which we identified to be important for microtubule bundling in prometaphase, is also present in the central spindle in anaphase, with an increased bundling activity[83,84]. Thus, it would be interesting to explore the relationship between the kinetochore-related mechanisms that promote the formation of microtubule bundles in prometaphase and anaphase, as well as their evolutionary aspects through comparison between different organisms.

We found a role of CENP-E, but not of Ndc80 and dynein, in overlap bundle formation. CENP-E, which is located in the fibrous corona connecting kinetochores to microtubules[67], is a very long and flexible molecule with a contour length of 230 nm[85]. This may allow

multiple CENP-E motors at an individual kinetochore to capture and bind laterally to several nearby microtubules[53,86]. In contrast, Ndc80 complex and cytoplasmic dynein are only 60 and 50 nm long when extended, respectively[87,88]. This difference in length may explain the importance of CENP-E in microtubule bundling, even though this it is not required for lateral attachment[89].

Given that CENP-E interacts with PRC1[83], their interaction may also be important for microtubule bundling. In addition to PRC1, other proteins that bind to antiparallel microtubules, such as Eg5, likely assist bundle formation. PRC1-interacting partners Kif4A and MKLP1 may also play a role, similarly to their roles in anaphase[27]. These hypotheses remain to be explored in future studies.

Bundling of microtubules by crosslinkers could in principle lead to bundling of most spindle microtubules into a single thick bundle. This is, however, not observed in animal spindles. In contrast, yeast spindles consist of a single microtubule bundle, which is likely due to a smaller number of microtubules, 40 in yeast versus 6000 in human spindles, and shorter spindles that prevent microtubule bending, 1.5 µm long in yeast versus 10 µm in human cells[38,90]. We propose that in animal spindles chromosomes crowding at the metaphase plate, together with coupling the bundle with kinetochore during its formation, promotes bundle separation by driving outward movements of the bundles, which results in spindle widening. Several lines of evidence from our work support this notion. During prometaphase, as more and more chromosomes arrive at the spindle midplane and fill it, the spindle gets progressively wider. In agreement with this, spindles with perturbed chromosome alignment, where chromosome arrangement resembles more the prometaphase state rather than metaphase, are narrower. Examples studied here include depletion of Ndc80, Kif18A, and CENP-E, which have different mechanistic origin of chromosome misalignment[74,75]. Thus, it is unlikely that the observed spindle narrowing is due to perturbation of specific functions of these proteins, but rather due to misalignment itself. Another argument comes from mitosis with unreplicated genome, in which kinetochores interact with spindle microtubules but the DNA is dispersed throughout the cells[10]. As expected, this resulted in narrow spindles, but fully formed bundles due to the presence of kinetochores.

Previous work has shown that in HeLa cells, which have a variable number of chromosomes[91], spindle width increases with the number of chromosomes[54]. This result also supports the chromosome crowding hypothesis. Finally, a striking example comes from the finding that a single microtubule bundle connects the spindle poles with chromosomes lying along the sides upon expression of a kinase-inactive Aurora B[92], in agreement with our hypothesis.

We have shown that PRC1-bound overlap bundles are required for the correction of synthetic kinetochore-microtubule attachments after spindle assembly from a monopole state. PRC1 depletion, which reduces the number of microtubules in the overalp bundles, resulted not only in an increase in lagging kinetochores, but also in decreased kinetochore phosphorylation. By combining these results with a previous work showing that Borealin binding to microtubules stimulates kinetochore phosphorylation and correction of erroneous kinetochore-microtubule attachments[71], we propose that Aurora B, being part of the Chromosome Passenger Complex together with Borealin, not only promotes the formation of PRC1-linked overlap bundles, but also uses them as tracks for the movement towards kinetochores to correct the wrong attachments. The molecular details of this PRC1-based error-correction mechanism remain an exciting topic for future studies.

In conclusion, our results reveal that a transition of microtubules from a loose array to bundles, dependent on kinetochores that serve as attraction sites and chromosomes that provide repulsion, promotes spindle assembly. In a broader sense, our study reveals general principles of organization of cytoskeletal filaments. The concepts of attraction sites as drivers of bundle formation may be relevant in other contexts, for example in the formation of actin bundles, where adherens junctions and focal adhesions may act as such sites. Overall, we expect that mechanisms similar to the ones identified here, though with different molecular players, act in a variety of cytoskeletal systems to generate specific filament arrangements required for their function.

## Methods

### Cell lines and culture

Experiments were carried out using unlabeled human HeLa-TDS cells (also referred to as untransfected HeLa cells) from the High-Throughput Technology Development Studio (Max Planck Institute of Molecular Cell Biology and Genetics, Dresden, Germany), HeLa-Kyoto BAC lines stably expressing PRC1-GFP (also referred to as HeLa PRC1-GFP cells) are courtesy of Ina Poser and Tony Hyman (Max Planck Institute of Molecular Cell Biology and Genetics, Dresden, Germany), and human hTERT-RPE1 (hTERT-immortalized retinal pigment epithelium) cells stably expressing CENP-A-GFP and Centrin1-GFP, courtesy of Alexey Khodjakov (Wadsworth Center, New York State Department of Health, Albany, NY, USA). All cell lines were cultured in flasks in Dulbecco's Modified Eagle's Medium with 1 g/L D-glucose, pyruvate and L-glutamine (DMEM, Lonza, Basel, Switzerland), supplemented with 10% (vol/vol) heat-inactivated Fetal Bovine Serum (FBS, Sigma-Aldrich, St. Louis, MO, USA) and penicillin (100 IU/mL)/streptomycin (100 mg/mL) solution (Lonza, Basel, Switzerland). For the selection of HeLa PRC1-GFP cell lines, 50 µg/ml geneticin was added to the medium (Life Technologies, Waltham, MA, USA). The cells were kept at 37 °C and 5% $CO_2$ in a humidified incubator (Galaxy 170 S $CO_2$, Eppendorf, Hamburg, Germany) and regularly passaged at the confluence of 70-80%. None of the cell lines were authenticated.

### Sample preparation, siRNA, plasmids, and dyes

At 80% confluence, DMEM medium was removed from the flask and cells were washed with 5 ml of phosphate buffered saline (PBS). Then, 1 ml 1% trypsin/ethylenediaminetetraacetic acid (EDTA, Biochrom AG, Berlin, Germany) was added to the flask and cells were incubated at 37 °C and 5% $CO_2$ in a humidified incubator for 5 min. After the incubation, trypsin was blocked by adding 2 ml of DMEM medium. For RNAi experiments, the cells were seeded to reach 60% confluence the next day and cultured on 35 mm uncoated dishes with 0.17 mm (#1.5 coverglass) glass thickness (MatTek Corporation, Ashland, MA, USA) in 2 mL DMEM medium with the supplements described above. After one day of growth, cells were transfected with either targeting or non-targeting siRNA constructs which were diluted in OPTI-MEM medium (Life Technologies, Waltham, MA, USA) to a final concentration of 100 nM in the medium with cells. All transfections were performed 48 h prior to imaging using Lipofectamine RNAiMAX Reagent (Life Technologies, Waltham, MA, USA) according to the instructions provided by the manufacturer. After four hours of treatment, the medium was changed to the DMEM medium. The constructs used were human HAUS6 siRNA (L-018372-01-0005, Dharmacon, Lafayette, CO, USA), human ON-TARGET plus SMART pool CENP-E siRNA (L-003252-00-0010, Dharmacon, Lafayette, CO, USA), ON-TARGET human CCDC99 siRNA (L-016970-00-0010, Dharmacon, Lafayette, CO, USA), human Ndc80 siRNA (HA12977117-004; Merck, Darmstadt, Germany), human ON-TARGET PRC1 siRNA (L-019491-00-0020, Dharmacon) and control siRNA (D-001810-10-05, Dharmacon, Lafayette, CO, USA).

For experiments with CENP-B-INCENP-GFP, unlabeled HeLa-TDS cells were transfected with CENP-B-INCENP-GFP (Addgene, plasmid #45238), and for live-cell imaging also with mCherry-PRC1 plasmid provided by Casper C. Hoogenraad (Utrecht University, Utrecht, The Netherlands). $1 \times 10^6$ cells were transfected using Lipofectamine 3000 Reagent (L3000001, Life Technologies, Waltham, MA, US) with 1.5 µg

of plasmid DNA for both plasmids. For CENP-E overexpression experiments, unlabeled HeLa-TDS cells were transfected with 2 μg of CENP-E-GFP plasmid (gift from Marin Barišić, Danish Cancer Society Research Center, Copenhagen, Denmark). To overexpress PRC1, unlabeled HeLa-TDS cells were transfected with 5 μg of mCherry-PRC1 plasmid. Transfection of both unlabeled HeLa-TDS and HeLa PRC1-GFP cells was performed 25–35 h before imaging.

In order to visualize microtubules, human hTERT-RPE1 cells stably expressing CENP-A-GFP and Centrin1-GFP and unlabeled HeLa-TDS cells were stained to a final concentration of 100 nM with a far-red silicon rhodamine (SiR)-tubulin-670 dye (Spirochrome, Stein am Rhein, Switzerland), 45 min to 2 h prior to imaging. In order to avoid dye efflux, a broad-spectrum efflux pump inhibitor verapamil (Spirochrome, Stein am Rhein, Switzerland) was added at a final concentration of 0.5 μM to the cells along with SiR-tubulin. For chromosome visualization, HeLa PRC1-GFP cells were stained with 100 nM SiR-DNA dye for 20 min to 2 h prior to imaging. To identify the cells in mitosis by condensed chromosomes in the PRC1 overexpression experiment on unlabeled HeLa-TDS cells, 50 μL of NucBlue Live Ready Probes Reagent (Hoechst 33342, Invitrogen by Thermo Fisher Scientific, Waltham, MA, USA) was added to the dishes 1 min before imaging.

For CENP-E Aurora B phosphosite experiments, stable U2OS cell lines with inducible expression of CENP-E-WT/T422A were generated by lentiviral infection and single clone selection. To replace the endogenous by exogenous CENP-E, cells were transfected with 3′UTR CENP-E siRNA (CCACUAGAGUUGAAAGAUA) 24 to 30 h prior fixation. Induction of exogenous expression was performed by the addition of 1 μg/ml doxycycline overnight (Sigma-Aldrich, St. Louis, MO, USA).

## Drug treatments
The stock solution of the Aurora B kinase inhibitor Barasertib (AZD1152-HQPA, Selleckchem, Munich, Germany) was prepared in Dimethyl sulfoxide (DMSO) to a final concentration of 1 mM. The working solution was prepared in DMEM at 600 nM and at the time of treatment, the working solution was added to cells at 1:1 volume ratio to obtain a final concentration of 300 nM; IC50 is 0.35 nM for this compound[93]. Barasertib was added at early prometaphase rosette in HeLa PRC1-GFP cell line. Appearance of predominantly prometaphase-like spindles[63] in the sample imaged 1 h post-treatment confirmed the effect of Barasertib on Aurora B.

The stock solution of the CENP-E inhibitor GSK-923295 (MedChemExpress, Monmouth Junction, NJ, USA) was prepared in DMSO at a final concentration of 8 mM. The working solution was prepared in DMEM at 200 nM. At the time of treatment, the working solution was added to cells at 1:1 volume ratio to obtain a final concentration of 100 nM; IC50 value of compound is 3.2 nM[94]. GSK-923295 was added at early prometaphase rosette in HeLa PRC1-GFP cell line immediately before the start of imaging, and 2 h before imaging in hTERT-RPE1 cell line expressing CenpA-GFP and Centrin1-GFP. Appearance of spindles blocked in prometaphase with a fraction of kinetochores trapped around polar region of the spindle at 30 min post-treatment[95] confirmed the effect of GSK-923295. For hTERT-RPE1 cells, mock treatment was the same concentration of DMSO that was used for preparation of the inhibitors.

Nocodazole (HY-13520, MedChemExpress, Monmouth Junction, NJ, USA) working solution was prepared in DMEM at 2 μM. At the time of treatment, the working solution was added to cells at 1:1 volume ratio to obtain a final concentration of 1 μM. Nocodazole was added at early prometaphase rosette in HeLa PRC1-GFP cell line immediately before the start of imaging.

Monastrol (HY-101071A/CS-6183, MedChemExpress, Monmouth Junction, NJ, USA) working solution (100 μM) was added in the dish with untransfected HeLa cells at a final concentration of 100 nM.

## Inducing mitosis with unreplicated genome (MUG)
MUG is induced by incubation of cells in hydroxyurea and caffeine[72], a gift from Anđela Horvat and Neda Slade (Laboratory for protein dynamics, Ruđer Bošković Institute, Zagreb, Croatia). The stock solutions of hydroxyurea and caffeine were prepared at a concentration of 175 mM and 200 mM, respectively. The stock solution was diluted in DMEM to the final concentration of 2 mM for hydroxyurea and 5 mM for caffeine. The amount of $1 \times 10^5$ HeLa PRC1-GFP cells were counted using the Improved Neubauer chamber (BRAND GMBH + CO KG, Wertheim, Germany), seeded and cultured in 2 ml DMEM at 37 °C and 5% CO₂ on 35 mm glass coverslip uncoated dishes with 0.17 mm (#1.5 coverglass) glass thickness (MatTek Corporation, Ashland, MA, USA). After 12 h, the medium was replaced with 2 mL DMEM containing 2 mM hydroxyurea and incubated for 20 h at 37 °C and 5% CO₂. After 20 h, the medium was again replaced with a medium containing 2 mM hydroxyurea and 5 mM caffeine and the cells were again incubated at 37 °C and 5% CO₂. The cells were imaged 18 h later.

## Immunocytochemistry and immunoblotting
HeLa cells expressing PRC1-GFP and untransfected HeLa cells were grown on glass-bottom dishes (14 mm, No. 1.5, MatTek Corporation, Darmstadt, Germany) and fixed by 1 ml of ice-cold methanol for 1 min at −20 °C. Following fixation, cells were washed 3 times for 5 min with 1 ml of PBS and permeabilized with 0.5% Triton-X-100 in water for 15 min at room temperature. To block unspecific binding, cells were incubated in 1 ml of blocking buffer (1% normal goat serum, NGS) for 1 h at room temperature. Cells were then washed 3 times for 5 min with 1 ml of PBS and incubated with 500 μl of primary antibody solution overnight at 4 °C. CENP-E cell lines were fixed with ice-cold methanol. Permeabilization and antibody incubation were performed using a blocking solution composed of 0.5% Triton, 5% FBS in PBS. The following primary antibodies were used: mouse monoclonal PRC1 (sc-376983, Santa Cruz Biotechnology), diluted 1:100, rat anti-alpha-tubulin YL1/2 (MA1-80017, Invitrogen, CA, SAD), diluted 1:500, mouse IgG monoclonal anti-GFP (Ref 11814460001, LOT42903200, Roche), diluted 1:100, rabbit anti-alpha-tubulin (SAB4500087, LOT 310379, Sigma-Aldrich), diluted 1:500, human anti-CREST (15-235, Antibodic sinc), 1:100, Rabbit anti-CENP-E (ab133583, Abcam, Cambridge, UK), diluted 1:500, Rabbit anti-Aurora B (ab239837, Abcam), diluted 1:100, anti-Hec1 (ab3613, Abcam), diluted 1:100, anti-Spindly (A301-354A, Biomolecules), diluted 1:100, anti-phospho-CENP-A (Ser7) (07-232, Sigma-Aldrich), diluted 1:500. After primary antibody, cells were washed in PBS and then incubated in 500 μL of secondary antibody solution for 45 min at room temperature. The following secondary antibodies were used: donkey anti-mouse IgG Alexa Fluor 594, diluted 1:250 (ab150112, Abcam, Cambridge, UK), donkey anti-mouse IgG Alexa Fluor 647, diluted 1:1000 (A31571, Invitrogen), donkey anti-rabbit IgG Alexa Fluor 647 (ab150075, Abcam), donkey anti-rat IgG Alexa Fluor 647 (ab150155, Abcam), goat anti-human IgG 594 (ab96909, Abcam), all diluted 1:500. Finally, cells were washed with 1 mL of PBS, 3 times for 10 min. This fixation was used for imaging HeLa PRC1-GFP cell line for siRNA silencing control, kinetochore and PRC1 visualization in vertical rosettes and for PRC1 visualization in different treatments (CENP-E overexpression, Aurora B relocalization) in untransfected HeLa cells.

To visualize alpha-tubulin in STED resolution in HeLa PRC1-GFP cell line, ice-cold methanol protocol was avoided because it destroyed the unstable fraction of microtubules[96]. Cells were washed with cell extraction buffer (CEB) and fixed by 3.2% paraformaldehyde (PFA) and 0.25% glutaraldehyde (GA) in PEM buffer (0.1 M PIPES, 0.001 M MgCl₂ × 6 H₂O, 0.001 M EDTA, 0.5% Triton-X-100) for 10 min at room temperature[97]. After fixation with PFA and GA, for quenching, cells were incubated in 1 mL of freshly prepared 0.1% borohydride in PBS for 7 min and after that in 1 mL of 100 mM NH₄Cl and 100 mM glycine in PBS for 10 min at room temperature. To block unspecific binding of antibodies, cells were incubated in 500 μL blocking/permeabilization

buffer (1% normal goat serum and 0.5% Triton-X-100 in water) for 2 h at room temperature. Cells were then incubated in 500 μL of primary antibody solution overnight at 4 °C. After the incubation with a primary antibody, cells were washed 3 times for 10 min with 1 ml of PBS and then incubated with 500 μl of secondary antibody for 2 h at room temperature. This protocol was used in all experiments where tubulin is visualized in STED mode. In all fixations, DAPI (1 μg/mL) was used for chromosome visualization.

For immunoblotting, cells were lysed in NP40 buffer (50 mM Tris–HCl pH 8, 150 mM NaCl, 5 mM EDTA, 0.5% NP-40, 1× EDTA-free protease inhibitor (Sigma-Aldrich), 1× phosphatase inhibitor cocktail (Sigma-Aldrich), 1 mM PMSF) and centrifuged at maximum speed for 15 min. Proteins were separated by a 5% SDS-PAGE gel and transferred to a nitrocellulose membrane (Bio-Rad). Membranes were incubated overnight with primary antibodies, mouse anti-CENP-E 1:500 (Santa Cruz Biotechnologies) and mouse anti-vinculin 1:5000 (Sigma-Aldrich), later incubated with HRP-conjugated secondary antibodies 1:10,000 (Jackson ImmunoResearch), and visualized by ECL (Bio-Rad).

## Imaging

STED microscope system (Abberior Instruments). STED microscopy was performed using the Expert Line easy3D STED microscope system (Abberior Instruments) with a 60×/1.2 UPLSAPO 60XW water objective (Olympus), avalanche photodiode (APD) detector, and Imspector software to acquire spindle cross-sections in horizontally and vertically oriented spindles in different stages of mitosis of HeLa cells expressing PRC1-GFP, immunostained for tubulin, with DNA stained by DAPI. STED images of tubulin were acquired in a single plane in the Alexa 594 channel with the excitation and depletion laser power at 35%, and pixel size set 20 nm. PRC1 and DNA were imaged in the same spindles in the confocal mode. Confocal mode was also used to image fixed prometaphase rosettes and metaphase spindles in untransfected and PRC1-GFP HeLa cells using eGFP and DAPI excitation lasers to visualize GFP and DAPI, respectively, or Alexa 488, 594, 647 excitation lasers depending on the secondary antibody. The laser power was 10%, except for Alexa 647 where it was 5%. Pixel size was 50 nm for prometaphase spindles and for imaging bridging fibers in metaphase, and 100 nm for metaphase spindles. Z-stacks of 41 focal planes, except in experiments on lagging kinetochores where the number of planes was 20, were acquired with 0.5 μm spacing to cover the whole spindle. For live-cell imaging of vertical metaphase spindles in HeLa PRC1-GFP cell line, the middle plane was recognized as the plane in which the bundles appear as clear dots of PRC1-GFP and the spindle cross-section is filled with chromosomes. Four planes close to the spindle midplane were acquired, separated by 0.5 μm. Pixel size was 50 nm. Laser power for eGFP excitation laser was 15%, and for SiR-DNA was 5%. Live-cell imaging of horizontal metaphase spindles was performed in HeLa PRC1-GFP cell line with Z-stacks of 41 planes using eGFP excitation light at 10% laser power and 5% laser power for SiR-DNA channel. Pixel size was 100 nm. The bundling assay was applied for imaging of tubulin with SiR-tubulin dye in HeLa PRC1-GFP cell line. Time-lapse images of the spindle midplane were acquired using only SiR excitation laser with 8% power. Time between repetitions was 1 min and pixel size was 100 nm.

Dragonfly spinning disk confocal microscope system (Andor Technology). To image the formation of PRC1-labeled bundles in HeLa cells expressing PRC1-GFP (bundling assay), confocal live-cell imaging was performed on a Dragonfly spinning disk microscope (Andor Technology, Belfast) using 63x/1.47 HC PL APO glycerol objective (Leica, Belfast) and Zyla 4.2 P scientific complementary metal oxide semiconductor (sCMOS) camera (Andor Technology). Images were acquired using Fusion software. During imaging, cells were maintained at 37 °C and 5% CO2 within a heating chamber (Okolab, Pozzuoli, NA, Italy). Only cells with vertically oriented spindles were imaged as follows: first a 41 plane Z-stack was acquired to cover the whole spindle using both 478 nm and 640 nm

excitation light at a 10% and 5% laser power, respectively. Then a time-series of images was required using only 478 nm excitation light. Every repetition recorded 9 middle frames of the vertical spindle, forming a stack of 9 slices separated 0.5 μm. Time between repetitions was 5.4 s. After the time-series, the second Z-stack was acquired, again using both excitation lasers. Total time-lapse movie was 9 min long. The same protocol was used for experiments with untreated cells, Haus6-depleted cells, and cells treated with Barasertib, GSK-923295, and nocodazole. For Ndc80, Haus6, and Spindly silencing control, a 41 plane Z-stack was acquired to cover the whole spindle using 478 nm, 620 nm and 408 nm excitation light at a 10% for first two and 5% laser power for DAPI channel. For all images pixel size was 100 nm.

Opterra confocal microscope system (Bruker). For counting the number of bundles and for bundle trajectories, HeLa cells expressing PRC1-GFP and hTERT-RPE1 cells expressing CENP-A-GFP and Centrin1-GFP were imaged using Bruker Opterra Multipoint Scanning Confocal Microscope (Bruker Nano Surfaces, Middleton, WI, USA). The system was mounted on a Nikon Ti-E inverted microscope equipped with a Nikon CFI Plan Apo VC 100x/1.4 numerical aperture oil objective (Nikon, Tokyo, Japan). The system was controlled with the Prairie View Imaging Software (Bruker). During imaging, cells were maintained at 37 °C in Okolab Cage Incubator (Okolab, Pozzuoli, NA, Italy). For optimal resolution and signal-to-noise ratio, 22 nm slit was used. For excitation of GFP, 488 nm diode laser line was used. The excitation light was separated from the emitted fluorescence by using Opterra Dichroic and Barrier Filter Set 405/488/561/640. Images were captured with an Evolve 512 Delta Electron Multiplying Charge Coupled Device (EMCCD) Camera (Photometrics, Tucson, AZ, USA) using 150-200 ms exposure times. For counting the number of bundles, Z-stacks of 41 focal planes were acquired with 0.5 μm spacing to cover the whole spindle. Horizontally oriented prometaphase spindles of HeLa PRC1-GFP cells with added SiR-DNA dye and hTERT-RPE1 cells expressing CENP-A-GFP and Centrin1-GFP with added SiR-tubulin dye were filmed every 10 and 5 min, respectively. Prometaphase cells were recognized by non-congressed chromosomes and relatively small spindle size. To image the dynamics of PRC1-labeled bundles in HeLa cells expressing PRC1-GFP, confocal live-cell imaging was performed. Only cells with vertically oriented spindles were imaged as follows: first a 41 plane Z-stack was acquired to cover the whole spindle using both 488 nm and 640 nm excitation light. Then a time-series of images was required using only 488 nm excitation light. Every repetition recorded 9 middle frames of the vertically oriented spindle, forming a stack of 9 slices separated 0.5 μm. Time between repetitions was 2.7 s. After the time-series, a second Z-stack was acquired, again using both excitation lasers. Total time-lapse movie was 4.5 min long. Laser powers were 10% and 5% for 488 nm and 640 nm excitation light, respectively.

CENP-E cell lines images were acquired using a Plan-Apochromat 63x/1.4 NA oil objective with a differential interference contrast mounted on an inverted Zeiss Axio Observer Z1 microscope (Marianas Imaging Workstation, 3i-Intelligent Imaging Innovations, Inc., Denver, CO, USA) equipped with an iXon Ultra 88 EM-CCD camera (Andor Technology, Belfast, UK). 0.5 μm separated Z-stacks were collected to cover the whole spindle.

## Image processing and data analysis

All images were analyzed in Fiji/ImageJ (National Institutes of Health, Bethesda, MD, USA). Raw images were used for quantification. Contrast was adjusted for clarity of presentation in the figures. MatLab (MathWorks, Natick, MA, USA) was used to create the plots, R studio (R Foundation for Statistical Computing, Vienna, Austria) to transform the horizontally oriented spindles into an end-on view. Figures were

assembled in Adobe Illustrator CS5 (Adobe Systems, Mountain View, CA, USA).

Bundling assay. For analysis of bundle formation, Squassh plugin in Fiji was used in the PRC1-GFP channel of the middle plane of the spindle. Segmentation was performed using Squassh[98]. In the main graphical user interface Squassh window, we set the parameter for background subtraction option, entering the window edge lengths in units of 14 pixels. Object detection was performed over the entire image. The higher value of the regularization parameter of 0.1 was used to avoid the segmenting noise-induced small intensity peaks. Intensity values were normalized between 0 for the smallest value occurring in the image, and 1 for the largest value. The sub-pixel segmentation was selected. The resolution of the segmentation was increased by an oversampling factor of 8 for 2D images. Local intensity estimation parameter was set to automatic. The Poisson model, recommended for confocal microscopes, was chosen. The following visualization options were selected: object intensities, number of objects and the mean object size in terms of area. Validation was performed by manually measuring object segmentation in the middle plane of a representative spindle.

Number of bundles. To count the bundles, horizontally oriented spindles were transformed into vertical (end-on) orientation using the code written in R programming language in R Studio (R Core Team., 2016). Before the transformation, the Z-stack of the spindle in a single channel was rotated in Fiji so that the spindle major axis was approximately parallel to the x-axis. The aberrations caused by refractive index mismatch between immersion oil and aqueous sample were taken into account in the program by multiplying Z-step size by a correction factor of 0.81 to obtain the correct Z-distance. This factor corresponds to the ratio of the cell diameter in y and z direction, assuming that a mitotic cell is spherical. Bundles were counted on the sum intensity images of 5 central planes of the transformed spindle. The number of bundles was determined by counting the bright spots using Multipoint tool in ImageJ.

Protein silencing. Analysis of the immunofluorescence signal was performed on the sum of all 41 planes. The territory of the spindle was encircled with a segmented line in Fiji and sum intensity was measured.

PRC1 intensity of early formed bundles in the prometaphase rosette. To measure the bundle intensity in vertically oriented early prometaphase rosette, bundles were encircled in Fiji and sum intensity was measured in only one plane of the cross-section where the bundle was of highest intensity. Bundles with bound kinetochores were measured in the plane where the kinetochore was visible close to the bundle. The brightest bundles (usually two) that connect the spindle poles in a straight line were not measured because they are formed before nuclear envelope breakdown. Thus, only bundles in the pro-metaphase network were measured.

PRC1 intensity of bundles in metaphase. To measure the bundle intensity in vertically oriented metaphase spindles, each bundle was encircled in Fiji with a fixed-size circle ($p = 0.61\,\mu m^2$) and mean intensity was measured in only one plane of the cross-section where DNA covered the whole cross-section. To calculate the PRC1 intensity of the bundle, mean intensity of PRC1 in the cytoplasm was subtracted from bundle PRC1 intensity. Further-more, to correct for the differences in PRC1 expression in different cells, the PRC1 bundle intensity was divided by the mean intensity of PRC1 in the cytoplasm to get the normalized PRC1 intensity.

PRC1 intensity of the network in early prometaphase. To measure the network intensity in vertically oriented prometaphase spindles, the region excluding the brightest bundles (the ones formed before nuclear envelope breakdown) was encircled in Fiji and mean intensity was measured in only one plane. As for the PRC1 intensity of bundles, the mean intensity of PRC1 in the cytoplasm was subtracted from the

network PRC1 intensity, and the result was divided by the mean intensity of PRC1 in the cytoplasm to get the normalized PRC1 intensity.

Tubulin intensity of bridging fibers in metaphase. Mean intensity of bridging fibers was measured on metaphase spindles using a line drawn along the bridging fiber using Line tool in Fiji. Only the region beneath the kinetochore pair was measured. Furthermore, to correct for the background, which was calculated as the mean intensity of tubulin in the cytoplasm next to the bridging fiber, and for differences in tubulin signal in different cells, tubulin bundle intensity was background subtracted and divided by the background to get the normalized tubulin intensity.

PRC1 line intensity in metaphase. In HeLa PRC1-GFP cell line, a line was drawn from pole to pole in a metaphase spindle, which was recognized by fully aligned chromosomes in control cells. In treatments that cause impaired chromosome congression, spindles with most chromosomes aligned were chosen. Line intensity plot was made by subtracting the cytoplasmatic PRC1 signal and dividing by it, providing the normalized PRC1 intensity.

PRC1 intensity in metaphase. For quantification of whole PRC1 intensity on metaphase spindles in immunofluorescent images we used sum of all planes composing a Z-stack. The spindle was encircled in Fiji and the total intensity of signal was calculated. From that signal, background signal of the same area was subtracted giving only PRC1 intensity of the spindle.

Aurora B-PRC1 correlation. Individual PRC1-labeled bridging fibers were encircled with a circle that goes from the top edge of the bundle to the bottom edge, and the sum intensity was measured. The same-size circle was used to measure the sum intensity of the Aurora B signal on the associated centromere. Normalized intensity of PRC1 and Aurora B was obtained by subtracting the cytoplasmatic mean intensity of the respective protein and dividing by it.

Phosphorylation of outer kinetochore. Sum intensity of CREST and CENP-A signal was measured in prometaphase cells. The spindle was boxed in a fixed square that was same for all cells. The intensity ratio was expressed by dividing the CENP-A-Ser7P sum intensity by the CREST sum intensity.

Spindle parameters. Spindle width and length were measured on horizontally oriented metaphase spindles using Line tool in Fiji.

Manual bundle tracking. The 2D trajectories of the bundles and their coordinates were tracked using Manual tracking tool to obtain bundle trajectories. The movement of all bundles was represented with respect to the brightest bundle(s) that appear before nuclear envelope breakdown and do not have chromosomes attached in early prometaphase, using Manual drift correction plugin in Fiji.

Fraction of DNA at the metaphase plate. Sum intensity of SiR-DNA signal from 41 planes was used. Fraction of DNA at the metaphase plate over time was measured as total intensity of the DNA within the line covering the equatorial region divided by the total intensity of all DNA in the cell. The line covering the equatorial region was defined in metaphase so that its width covered 10% of the metaphase spindle length in the last time frame. Both length and width of the line were fixed based on the last time frame and used in every time frame.

Lagging kinetochores were identified by the CREST signal together with the DNA signal in the central part of the spindle in anaphase, between the two segregated groups of kinetochores/chromosomes.

## Theoretical model

To describe the dynamical transition from array like distribution of microtubules to well organized multiple bundles during pro-metaphase, we consider both equilibrium and nonequilibrium processes involving microtubules, crosslinking proteins,

chromosomes, and kinetochores. The model we construct is placed in an one-dimensional geometry with spatial coordinate $x$, taken from the part of the circumference of the disc, $L$, that is obtained by taking a vertical cross-section at the midplane of a prometaphase spindle as shown in Fig. 5a. We choose a one-dimensional geometry as it substantially reduces the complexity of the problem. One-dimensional geometry preserves the major features of the system that we study because in early prometaphase the kinetochores are predominately distributed along a line that forms a circle on the periphery of the prometaphase spindle (see Fig. 2b).

In our model, we include interactions between microtubules, crosslinkers and kinetochores. Microtubules in the vertical cross-section appear as point like objects and their distribution is described by density $\rho$. In order to calculate microtubule distributions, we construct a Landau–Ginzburg free energy, $F$, as

$$\beta F[\rho] = \oint_{\text{entire L}} dx \left\{ \frac{w}{4}\rho^4 + \frac{\kappa}{2}\left(\frac{\partial \rho}{\partial x}\right)^2 - \zeta\rho\psi\rho - \sum_{i=0}^{N-1}\alpha\rho^2\delta[x-(2i-1)d] \right\},$$
(1)

where an inverse of Boltzmann constant multiplied by the temperature is denoted by $\beta = (k_B T)^{-1}$. This free energy incorporates several types of interactions between microtubules, crosslinkers, and kinetochores. In the first term we describe local microtubule repulsion as nonlinear interaction with strength $w$. The second term captures a non-local repulsion of strength $\kappa$, which is coming from interaction between chromosomes and microtubules. The third term describes an attractive interaction between two microtubules mediated by crosslinkers and with coupling constant $\zeta$. The last term represents attraction of strength $\alpha$ between individual kinetorchores and microtubules, where $N$ is the total number of kinetochores. This local attraction is described by Dirac delta function. In our system, kinetochores are placed in regular intervals of given integer index $i$, forming lattice-like structure with spacing $2d = L/N$. For simplicity, we assume that crosslinkers are uniformly distributed with a constant value $\psi$.

As the density conservation is in place, we proceed to write a conserved dynamics of the field $\rho$,

$$\frac{\partial \rho}{\partial t} = -\frac{\partial J}{\partial x}$$
(2)

where the current is given by $J = -M_\rho \partial\mu/\partial x$. The chemical potential is obtained by minimizing free energy, $\mu = \frac{\delta}{\delta\rho}(\beta F)$. A constant $M_\rho$ denotes mobility of microtubules and in general it is related directly to the diffusivity of the microtubules. Thus, considering the equilibrium properties of the system we obtain the time evolution as,

$$\frac{\partial \rho}{\partial t} = M_\rho \left\{ w\frac{d^2\rho^3}{dx^2} - \kappa\frac{d^4\rho}{dx^4} - 2\zeta\psi\frac{d^2\rho}{dx^2} - 2\alpha\frac{d^2}{dx^2}\sum_{i=0}^{N-1}\rho\delta(x-(2i-1)d) \right\}$$
(3)

Moreover, rapid polymerization and depolymerization of microtubule filaments causes the processes of creation and annihilation of those filaments in the mid plane of the vertically oriented spindle. We assume that microtubules are created with a constant rate $\omega_{\text{on}}$ and get disassembled with a rate $\omega_{\text{off}}$. Finally, we calculate how microtubule density changes in time $t$ by incorporating this nonequilibrium turnover dynamics of microtubules along

with the equilibrium processes as,

$$\frac{\partial \rho}{\partial t} = M_\rho \left\{ w\frac{d^2\rho^3}{dx^2} - \kappa\frac{d^4\rho}{dx^4} - 2\zeta\psi\frac{d^2\rho}{dx^2} \right.$$
$$\left. - 2\alpha\frac{d^2}{dx^2}\sum_{i=0}^{N-1}\rho\delta(x-(2i-1)d) \right\} + \bar{\omega}_{\text{on}} - \omega_{\text{off}}\rho.$$
(4)

Nucleation rate per unit length is given by $\bar{\omega}_{\text{on}} = \frac{\omega_{\text{on}}}{L}$.

Numerical solution of the steady-state nonlinear differential equation by power series expansion: The steady-state density $\rho_f$ is obtained by setting $\frac{\partial\rho_f}{\partial t} = 0$ in Eq. 4 as,

$$M_\rho \left\{ w\frac{d^2\rho_f^3}{dx^2} - \kappa\frac{d^4\rho_f}{dx^4} - 2\zeta\psi\frac{d^2\rho_f}{dx^2} - 2\alpha\frac{d^2}{dx^2}\sum_{i=0}^{N-1}\rho_f\delta(x-(2i-1)d) \right\}$$
$$+ \bar{\omega}_{\text{on}} - \omega_{\text{off}}\rho_f = 0$$
(5)

This is an ordinary nonlinear differential equation with the highest order 4 and with cubic nonlinearity. To solve it, we use a power series expansion given by

$$\rho_f(x) = \sum_{j=0}^{\infty}\frac{a_j}{j!}x^j$$
(6)

where $a_j$ are the coefficients of the expansion. Because of the non-linearity of Eq. 5, the expression of higher order terms becomes more complex as it couples a larger number of different coefficients in the expansion. Thus, we search for an approximate solution which can be calculated by several lowest orders of the expansion.

The system consists of periodically placed attractive $\delta$ functions which has a mirror symmetry with respect to $x = 0$. Thus, we search for solutions that have a mirror symmetry, $\rho_f(-x) = \rho_f(x)$. This condition implies that in Eq. 6 only even powers in the expansion will survive. To find an approximative solution, we truncate Eq. 6 until 6th order and the expansions are given by

$$\rho_f(x) = a_0 + \frac{a_2}{2!}x^2 + \frac{a_4}{4!}x^4 + \frac{a_6}{6!}x^6$$
(7)

By inserting this expansion into Eq. 5 we obtain a set of recursive relations between coefficients that read as,

$$a_4 = \frac{1}{\kappa M_\rho}\left(3wM_\rho a_0^2 a_2 - 2\zeta\psi M_\rho a_2 + \bar{\omega}_{\text{on}} - \omega_{\text{off}}a_0\right)$$
(8)

$$a_6 = \frac{2}{\kappa M_\rho}\left[wM_\rho\left(9a_0 a_2^2 + \frac{3}{2}a_0^2 a_4\right) - \zeta\psi M_\rho a_4 - \frac{\omega_{\text{off}}}{2}a_2\right]$$
(9)

Proceeding in the same way one can obtain higher order corrections into the microtubule density profile $\rho_f$.

To fix the two remaining integration constants, $a_0$ and $a_2$, we use the boundary conditions. The boundary condition that involves the Dirac delta function, which reflects as a discontinuity in the first derivative of $\rho_f$ at $\pm d$,

$$\kappa\left[\frac{d\rho_f}{dx}\Big|_{x=-d+\epsilon} - \frac{d\rho_f}{dx}\Big|_{x=d-\epsilon}\right] = -2\alpha\rho_f(d),$$
(10)

in the limit $\epsilon \to 0$. The other boundary condition takes into account conservation of microtubule numbers, which reads

$$\int_{-d}^{d} \rho_f(x)dx = \frac{2d\bar{\omega}_{on}}{\omega_{off}}, \qquad (11)$$

By inserting expansion given by Eq. 7 and expression from Eqs. 10, 11, we fix the two remaining integration constants, $a_0$ and $a_2$.

We tested the validity of this approach by comparing the result from 4th order with 6th order and found that the difference is around 5% for d = $2\,\mu m$ and $\alpha = 10\,\mu m^2$, suggesting that numerical error is below 5%. This error reduces to 0.5% when the inter-kinetochore distance is $1\,\mu m$, keeping all other parameters fixed.

Effective free energy of the system: Once the microtubule density profile $\rho_f$ is obtained, we construct an effective free energy description by putting back $\rho_f$ into Eq. 1. As the time-dependent Ginzburg-Landau equation contains nonequilibrium contribution of microtubule turnover that also governs the shape of the density profile, we use the term effective free energy, instead of free energy. We use this approximate free energy description of the microtubule-crosslinker-kinetochore system to calculate the stability of a uniform lattice spacing.

To check the stability of the system, we compare the effective free energy of a lattice with a uniform lattice spacing and that with a non-uniform spacing in the system consisting of two kinetochores. The lattice stretches from $-2d$ to $+2d$ with total length $L = 4d$ and a periodic boundary condition is imposed on it. In the uniformly spaced lattice, kinetochores are at positions $\pm d$, yielding the distance between kinetochores $2d$. In the deformed lattice kinetochores are placed at positions $-d + \epsilon$ and $d - \epsilon$, with two different distances between kinetochores, $2d - 2\epsilon$ and $2d + 2\epsilon$, where $\epsilon$ denotes small displacement from the uniform lattice. Thus, the effective free energy of the deformed lattice is given by,

$$\beta F_{eff} = \int_{-2d}^{2d} dx \left\{ \frac{w}{4}\rho_f^4 + \frac{\kappa}{2}\left(\frac{\partial \rho_f}{\partial x}\right)^2 - \zeta \rho_f \psi \rho_f \right.$$
$$\left. - \alpha \rho_f^2 \left[\delta(x + d - \epsilon) + \delta(x - d + \epsilon)\right] \right\}, \qquad (12)$$

We evaluate this integral by splitting the integration interval onto three subintervals: $[-2d, -d + \epsilon]$, $[-d + \epsilon, d - \epsilon]$, and $[d - \epsilon, 2d]$. Approximate solutions in these intervals can be written as $\rho_f = \rho_0 \pm \Delta\rho$, where $\rho_0$ denotes microtubule density for the uniformly spaced lattice and $\Delta\rho$ denotes a small distortion from microtubule density of the uniformly spaced lattice. The plus sign corresponds to the first and the third intervals, and the minus sign corresponds to the second interval.

The lowest correction term is of the order of $\epsilon^2$. After evaluating Eq. 12, we find that the only nontrivial contribution comes from terms that represent kinetochore-microtubule attraction, $\delta$ functions, yielding the difference in the effective free energy $\beta\Delta F_{eff} = [2\alpha\rho_0(d^2\rho_0/dx^2)]_{x=d}\,\epsilon^2$. For the parameters explored in Fig. 5, the microtubule density $\rho_0$ and its second derivative have positive values. Thus, the effective free energy difference has a positive value for a small displacement of kinetochores from the uniform lattice. This result indicates that the evenly spaced kinetochore lattice is placed at the local minimum of effective free energy and that for this choice our system is in stable equilibrium.

Connections between model parameters and biological processes: The parameters $\omega_{on}$ and $\omega_{off}$ are related to microtubule polymerization and depolymerization respectively, across the circumferential ring of the midplane cross-section. To calculate $\omega_{on}$ we first estimate the current of plus-ends of growing microtubules, $J_c = \rho_c v_g$, from the linear density of the microtubule in the midzone, $\rho_c$, and the microtubule growth velocity, $v_g$. Previous measurements of

the EB1 comets that are passing through the spindle midplane of thickness of $0.5\,\mu m$ reveal that around 27000 spots were present in the zone during 75 frames in 3 cells, yielding the $\rho_c = 27000/(0.5 \times 75 \times 3)\,\mu m^{-1}$ (Fig. 9b from ref. 99). From the same measurements, the growth velocity was $v_g = 0.465\,\mu m \cdot s^{-1}$ (Fig. 6a from ref. 99), yielding $J_c = 110\,s^{-1}$. The circumferential ring of the midplane cross-section has radius $5\mu m$, but the area where kinetochores lay is $1\,\mu m$ thick and it represents approximately 1/3 of the midplane cross sectional area, giving rise $\omega_{on} = J_c/3 = 40\,s^{-1}$. The other parameter, $\omega_{off}$, is estimated from the total number of microtubules in the spindle, $N = J_c/\omega_{off} = 6300$, based on the electron tomography data[38], giving $\omega_{off} = 0.02\,s^{-1}$.

The parameter $\zeta$ is related to the binding energy of one PRC1 molecule that crosslinks a pair of microtubules. We estimate binding energy of a single PRC1 as a probability of two microtubules to get close enough so that a single PRC1 can crosslink them as $(\rho x_{PRC1})^2$, which is multiplied by a binding energy of that microtubule pair formation, $e$. Thus, the parameter related to the binding energy has a value $\zeta = \beta e x_{PRC1}^2$, where $x_{PRC1}$ denotes the length of a PRC1 dimer. We estimate the binding energy, $e$, to the order of $10k_BT$ as it should be strong enough to resist thermal fluctuations and at the same time must be weak enough so that PRC1 can undergo spontaneous dissociation as seen experimentally[29]. The unstretched length of the PRC1 dimer $x_{PRC1} = 40\,nm$, based on ref. 100, yielding $\zeta = 0.016\,\mu m^2$.

The kinetochore-microtubule coupling strength, $\alpha$, is associated with the binding energy of the microtubules to the kinetochore. We estimate it as the coupling strength between a single microtubule and kinetochore, $\bar{e}$, multiplied by the number of available binding, $n_{site}$, and normalized by the saturating microtubule surface density at the binding sites, $\rho_{site}$, yeilding $\alpha = n_{site}\beta\bar{e}/\rho_{site}$. We estimate the binding energy per microtubule to be $\bar{e} = 10k_BT$, which is similar to the binding energy of a single motor protein to the microtubule[101], the number of available binding sites $n_{site} = 10$ is estimated from the number of microtubules bound to one kinetochore[38], and the saturating microtubule surface density is $\rho_{site} = 80\,\mu m^{-2}$ (ref. 38), yielding $\alpha = 1.25\,\mu m^2$.

## Statistics and reproducibility
Data are given as mean ± s.e.m., unless otherwise stated. Means of two groups were compared by Student's t-test (two-tailed and two-sample unequal-variance), and means of more than two groups by one-way ANOVA test and Tukey's HSD post hoc test, $p < 0.05$ was considered statistically significant. The numbers of microtubule bundles, cells, and independent experiments are given in the figure captions. In cases where representative spindle images are shown, similar observations were made in at least 10 spindles from at least 3 independent experiments. Graphs were generated in Matlab (MathWorks, Natick, MA, USA). Fiji was used to scale images and adjust brightness and contrast. Figures were assembled in Illustrator (Adobe Systems, Mountain View, CA, USA).

## Reporting summary
Further information on research design is available in the Nature Portfolio Reporting Summary linked to this article.

## Data availability
All relevant data supporting the key findings of this study are available within the article and its Supplementary Information files. All raw images used in this work are available upon reasonable request. Source data are provided with this paper.

## Code availability
The code for the theoretical model of bundle formation is available at https://github.com/subhadip-physics/microtubule-bundling.git.

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

## Acknowledgements

We thank Ina Poser, Tony Hyman, Alexey Khodjakov, Casper Hoogenraad, and Neda Slade for cells, plasmids, and chemicals, Juraj Simunić and Martina Manenica for help with pilot experiments, Ivana Šarić for the drawings and assembling the figures, and all members of Tolić and Pavin groups for helpful discussions. This work was funded by the Croatian Science Foundation (HRZZ, project PZS-2019-02-7653 granted to I.M.T. and M.B.), the European Research Council (ERC Synergy Grant, GA Number 855158, granted to I.M.T. and N.P.), and projects co-financed by the Croatian Government and European Union through the European Regional Development Fund—the Competitiveness and Cohesion Operational Program: IPSted (grant KK.01.1.1.04.0057) and QuantiXLie Center of Excellence (grant KK.01.1.1.01.0004).

## Author contributions

I.M.T., M.B., and N.P. conceived the project. J.M. performed and analyzed all experiments, except those with lagging kinetochores, PRC1 and CENP-E overexpression, anaphase entry, and a subset of immunostainings, which were done by M.Ć., and experiments on T422A CENP-E mutant, which were done by S.E. S.G. developed the theoretical model. I.M.T. and N.P. supervised the experimental and theoretical work, respectively. I.M.T., J.M., and N.P. wrote the paper with input from all authors.

## Competing interests

The authors declare no competing interests.
