## [Peer Review File · Nature Communications]

Kinetochore- and chromosome-driven transition of microtubules into bundles promotes spindle assemblyREVIEWER COMMENTS

Reviewer #1 (Remarks to the Author):

A major advance in our understanding of cell division machinery was the realization that a large number of microtubules within the mitotic spindle must organize into antiparallel bundles. The number of these bundles, their distribution, mechanical properties, and dynamics must be properly controlled for the spindle to be fully functional. Yet, how these controls are enacted, what determines when and how the bundles form and integrated into the spindle is not fully understood. This manuscript provides additional insights into the dynamics of microtubule bundling during early stages of mitosis. Further, several factors that perturb the number of bundles, dynamics of their assembly, as well as the number of microtubules and the amount of crosslinkers within a typical bundle are identified via perturbation of various proteins. The candidates for inhibition analyses are chosen intelligently based on the previously identified functions of these proteins in the formation and/or maintenance of proper connections between the chromosomes and spindle microtubules. Several novel and interesting observations emerge from the work described in the manuscript. Most data are of high quality and the experimental results are supported by computational modeling. The manuscript is good contribution to the field of cell division. Although, this review lists several issues that require the authors' attention, these issues can be easily addressed in the revision. The only real limitation of this study, in this reviewer's opinion, is that only a modest insight into the molecular mechanism of bundle formation is offered. None of the presented experiments reveals a specific and well-defined function for CenpE (or Aurora B) in the formation of antiparallel bundles. Instead, the work establishes that various perturbations of spindle assembly are accompanied by abnormalities in the dynamics of microtubule bundling. This is an interesting and important finding, yet it does not provide sufficient proof for the scenario presented in Fig.7.

Below are some specific comments that I hope will help the authors to strengthen the manuscript.

Some problematic pieces of data:

The cell shown in Video S1, Figs.1B, and S1C-F appears to be photodamaged. A 10-min long recording initiated in early prometaphase should cover the entire process of spindle assembly through early metaphase. However, morphology of the spindle during later timepoints of the movie is clearly abnormal. The shape of the spindle is quite different from the metaphase panel in Fig.1A – it's not circular, the perimeter is continuous, and the distribution of bundles is not nearly as uniform as in the fixed-cell image. Thus, dynamics of bundle formation might not be quite normal. A different example of live recordings should be presented.

There appears to be conflicting data on the dynamics of PRC1 segment size in untreated cells. This metric decreases linearly from ~6 to ~1 μm^2 in Fig. S1D; however, the 'control' curve in Fig.S1K is flat at ~1 μm^2 throughout the recording. This needs to be explained.

Fig.S4 C and D. The images raise concerns because the controls look so abnormal. In panel C, the spindle isn't even bipolar in the control cell – the image appears to illustrate a failure in spindle assembly. In D, the control spindle is way too short for a normal cell. Further, the quality of CenpE staining is unacceptable. CenpE antibodies are famous for the clarity of kinetochore staining, indeed, the beautiful crescent of prometaphase kinetochore stained for CenpE grace quite a few journal covers. I am really surprised to see such a high speckled background and lack of clearly delineated kinetochores in a mock-depleted cell stained for CenpE.

Line 197. This formulation is an example of the tendency to formulate conclusions quite strongly. The Aurora inhibition results reveal a significant decrease in PRC1 and tubulin density, within microtubule bundles. This observation does not prove that "Aurora B activity regulates the number of microtubules within the bridging fibers in metaphase". Not every bundle is a bridging fiber and disintegration of the bundles in Aurora B inhibited cells is evidence of specific regulation of microtubule numbers.

Fig.2 F and G. The measurements are presented (and later interpreted) as "mean tubulin intensity in

the middle of the bridging fiber in control and Aurora B-inhibited cells". Yet, no bridging fiber is apparent in the bottom image of panel F. The numbers that are interpreted as a decrease of MT numbers within bridging fibers but the image suggests that a lower number of centromeres associate with bridging fibers.

Fig.2 I. Approximately 1.5 fold increase in the overall PRC1 density measured via line-scans in cells with displaced Aurora B is puzzling as the bundles are dimmer in Aurora B-inhibited cells. This should be discussed.

Fig.3B, C. The same problem as in Fig.2 F,G – there is no bridging fiber associated with this centromere. The image depicts two K-fibers in the classic amphitelic arrangement.

Minor stylistic points

Figure panels are not referenced sequentially. Specifically, the text progresses from Fig.2C to Fig.3, then to the supplementary Fig.3 and back to Fig.2.

'Squassh' is spelled as 'squash' throughout the manuscript.

Lines 49-53: There is no proof that effects of Eg5 inhibition are directly linked to the function of this protein within microtubule bundles. Eg5 acts on antiparallel individual microtubules as well. Indeed, references 13-16 do not implicate overlap bundles in the Eg5 phenotype.

Fig.1A. The images depict cells in early prometaphase, late prometaphase, and metaphase. They should be labeled as such. Congression is a process not a stage of mitosis. Further, judging by the width of chromosome spread in axial direction, similar numbers of chromosomes are 'congressed' (reside near the spindle equator) in the depicted cells.

Line 179. This statement references two publication that reached somewhat different conclusions. Here is a quote from ref.54 regarding ref.53: "Thus, the association of non-KMT bundles solely with one pair of sister K-fibers, which has been described in the light microscope (Polak et al., 2017), is not confirmed in ET reconstructions...". Not a big deal but looks somewhat awkward. Indeed, a more straightforward segue to focusing the chapter on the interplay of kinetochores and PRC1 bundles would be a reference to the recent work by Renda and co-authors (2022) where interactions between kinetochores and bundles during early prometaphase were directly observed.

Reviewer #2 (Remarks to the Author):

In this study, Matkovic et al. study the formation of overlap bundles of microtubules, comprised of antiparallel microtubules extending from opposite spindle halves, within mitotic spindles in human tissue cultured cells. By using STED super-resolution microscopy, the authors show that a transition from a diffuse array of microtubules to a bundled organization occurs between early prometaphase and metaphase. This progressive appearance of bundles is concomitant to chromosome alignment, and is accompanied by increased spacing between bundles and their decoration by the microtubule cross-linker PRC1. Functional analysis revealed that the initial array of microtubules is largely nucleated by the augmin complex. Individual microtubules are then initially bundled by laterally interacting with CENP-E at kinetochores, under the control of the AuroraB kinase, before being cross-linked by PRC1. Importantly, microtubule bundling is independent of Ndc80, and thus of kinetochore fiber formation. By depleting various microtubule motors or kinetochore components, the authors show that microtubule bundles are progressively separated from each other by the steric interactions induced by chromosome alignment on the metaphase plate. They also provide a theoretical model in support of their in experimental findings. Finally, to test the functional relevance of overlap bundles for chromosome segregation, the authors reduced their assembly (by about 50%) by depleting PRC1,

and analyzed error correction in cells washed-out of monastrol. This revealed a mild but significant increase in the fraction of lagging kinetochores in cells with decreased overlap bundles (from 2% to 11%). Overall, this is an interesting study, which highlights the mechanism of assembly of an understudied population of microtubule bundles in mitotic spindles, and also provides a potential function for them during chromosome alignment. Listed below are suggestions and relatively minor points that would improve the final version of this manuscript:

- 1-Since the authors propose that CENP-E activity is regulated by AuroraB for initial microtubule bundling, it would be interesting to include analysis of the non-phosphorylatable T422A CENP-E mutant (from Kim, Holland et al., Cell, 2010).
- 2-To further confirm that overlap bundles can serve as tracks for the CPC to phosphorylate kinetochore targets, analysis of other phospho-targets (Knl1, Dsn1, Ndc80 or else) could be included in a revised version of this manuscript.
- 3-Include Extended Fig. 3H and 3J in Fig. 2 and 3 respectively.
- 4-Add a control panel in Fig. 3B.
- 5-Add images of siKif18A in Fig. 4.
- 6-Line 453: edit sentence.
- 7-Line 518: edit 'thought'.
- 8-Line 573: edit 'combing'.

Reviewer #3 (Remarks to the Author):

This contribution hypothesizes a mechanism for mitotic spindle formation through formation of microtubule bundles mediated by competing attraction/repulsion mechanisms involving kinetochores and chromosomes.

An experimental part using various staining and chromosome modification techniques (e.g., MUG, Ndc80 depletion) attempts to support the hypothesis. Ascribing observations to specific modifications of the system is difficult in the absence of a model of the biomechanics.

The authors introduce a simple one-dimensional model purporting to validate the spindle formation hypothesis. Within the paper itself, the model is perfunctorily described, and later in the Discussion section stated to "have shown" (l. 483) the hypothesis. While fully appreciating the difficulty of constructing mathematical models of the biomechanics that are both insightful and sufficiently simple, a haphazard and careless presentation of the mathematical model does not lend credence to the overall conclusions within this contribution.

In supplementary Theoretical model section, some approaches are questionable, the style of presentation tends to obfuscation, and errors are present in the derivation:

(1) A one-dimensional model along the circumference of the spindle midplane is introduced (Fig. 5), and a partial differential equation is introduced expressing microtubule conservation using a common time dependent Ginzburg-Landau model:

$$\rho_t = M \operatorname{div}(\operatorname{grad}(F))$$

using a hypothesized free energy model containing ρ^4 , ρ^2 , and $(\rho_x)^2$ terms. The choice of a simple proportionality by the mobility M should be discussed, since there are various ways to introduce time-dependence in a Ginzburg-Landau model. The two-dimensional model is reduced to one dimension along the circumferential variable x , valid if the region of interest is thin by comparison to the mean radius. Fig. 4B suggests otherwise: the spindle mid-plane ring is not of negligible thickness, hence some discussion of the two-dimensional model is warranted.

(2) The model contains five independent parameters w , κ , ζ , α , ω . Non-dimensional values are given in the discussion of Fig. 5 with reference to the Methods. No information is available

within the Methods section on why these particular parameters were chosen. Absent some link between the chosen model parameters and the biophysics of the system, only qualitative behavior conclusions can be drawn from the theoretical model.

(3) The free energy model itself contains fixed kinetochore positions (the Dirac delta terms). Since these positions are fixed, it is not surprising that model parameters can be chosen to show "bundling" or, in mathematical terms, attraction points - they were built into the model from the start. This qualitative behavior should be completed by discussion of the model parameter space, and of the case in which kinetochore positions themselves change in time (as suggested by, e.g., Fig. 3, F).

(4) The model itself contains annoying if minor errors:

(a) kinetochores are equally-spaced along the circumference, but the free energy contains only a contribution from that of indices $\pm(2n+1)$. Are only nearest neighbors considered, or is a sum missing?

(b) There is an extra M_{ρ} factor term in the third right-hand term of the unnumbered equation above equation (2)

(c) The non-dimensionalization is incorrect. Using the stated reference quantities the coefficient of the first right-hand term of (2) becomes

$$\omega_{\text{off}} M \tilde{w}^{4/3}$$

instead of $\tilde{w} M$ as in the unnumbered equation above.

(5) The non-linear time-dependent Landau-Ginzburg model is subsequently linearized and treated in an unstated nearest-neighbor hypothesis. The elementary and tedious discussion of the solution of a linear fourth-order ODE can be streamlined, and more effort dedicated to determining the range of validity of the linearization. If this were a small perturbation around the bundled state, the model might be valid. But the whole point of the model is to support the hypothesis of evolution from a uniform microtubule distribution to one concentrated at the fixed kinetochore locations. Nonlinear dynamical systems can exhibit complicated phase space trajectories in this case.

RESPONSE TO REVIEWERS

Kinetochores- and chromosome-driven transition of microtubules into bundles promotes spindle assembly

We thank the reviewers for providing perceptive comments on our manuscript. We have carefully thought about each point, and are outlining our revisions of the paper, with reviewer comments in black and our answers in blue. In the manuscript file, all changes are marked in blue.

REVIEWER 1:

Reviewer #1: A major advance in our understanding of cell division machinery was the realization that a large number of microtubules within the mitotic spindle must organize into antiparallel bundles. The number of these bundles, their distribution, mechanical properties, and dynamics must be properly controlled for the spindle to be fully functional. Yet, how these controls are enacted, what determines when and how the bundles form and integrate into the spindle is not fully understood. This manuscript provides additional insights into the dynamics of microtubule bundling during early stages of mitosis. Further, several factors that perturb the number of bundles, dynamics of their assembly, as well as the number of microtubules and the amount of crosslinkers within a typical bundle are identified via perturbation of various proteins. The candidates for inhibition analyses are chosen intelligently based on the previously identified functions of these proteins in the formation and/or maintenance of proper connections between the chromosomes and spindle microtubules. Several novel and interesting observations emerge from the work described in the manuscript. Most data are of high quality and the experimental results are supported by computational modeling. The manuscript is a good contribution to the field of cell division. Although, this review lists several issues that require the authors' attention, these issues can be easily addressed in the revision. The only real limitation of this study, in this reviewer's opinion, is that only a modest insight into the molecular mechanism of bundle formation is offered. None of the presented experiments reveals a specific and well-defined function for CenpE (or Aurora B) in the formation of antiparallel bundles. Instead, the work establishes that various perturbations of spindle assembly are accompanied by abnormalities in the dynamics of microtubule bundling. This is an interesting and important finding, yet it does not provide sufficient proof for the scenario presented in Fig.7.

Below are some specific comments that I hope will help the authors to strengthen the manuscript.

Some problematic pieces of data:

The cell shown in Video S1, Figs.1B, and S1C-F appears to be photodamaged. A 10-min long recording initiated in early prometaphase should cover the entire process of spindle assembly

through early metaphase. However, morphology of the spindle during later timepoints of the movie is clearly abnormal. The shape of the spindle is quite different from the metaphase panel in Fig. 1A – it's not circular, the perimeter is continuous, and the distribution of bundles is not nearly as uniform as in the fixed-cell image. Thus, dynamics of bundle formation might not be quite normal. A different example of live recordings should be presented.

We thank the reviewer for the careful reading of our manuscript and for raising this issue. In the original manuscript, we showed the cell from Video S1, Figs. 1B, and S1C-F only until 7 minutes, in order to focus on early bundle formation. To address the issue of photodamage, we now include the movie of this cell until 9 minutes in Video S1, which shows that the spindle cross-section is becoming circular, and the distribution of the bundles is becoming uniform, suggesting normal spindle assembly. Additionally, the same cell is shown in Fig. S1B bottom left, where it is visible that the chromosomes have entered the central part of the spindle cross-section at 9 minutes, indicating that the cell is at a late prometaphase or early metaphase stage at that time point. Moreover, we added another cell, as suggested by the referee, in Video S1 (on the right), Fig. S1B (right column) and S1C, which reached metaphase within 9 minutes. It is important to mention that these HeLa cells typically reach metaphase around 10 minutes or even later, which can be seen from Fig. 4d, where the cells were imaged at 1-minute intervals and thus photodamage is unlikely.

There appears to be conflicting data on the dynamics of PRC1 segment size in untreated cells. This metrics decreases linearly from ~6 to ~1 μm^2 in Fig. S1D; however, the 'control' curve in Fig. S1K is flat at ~1 μm^2 throughout the recording. This needs to be explained.

We thank the reviewer for pointing out this error, which is now corrected, i.e., the curves for Control and Haus siRNA were swapped in Supplementary Fig 11.

Fig. S4 C and D. The images raise concerns because the controls look so abnormal. In panel C, the spindle isn't even bipolar in the control cell – the image appears to illustrate a failure in spindle assembly. In D, the control spindle is way too short for a normal cell. Further, the quality of CenpE staining is unacceptable. CenpE antibodies are famous for the clarity of kinetochore staining, indeed, the beautiful crescent of prometaphase kinetochore stained for CenpE grace quite a few journal covers. I am really surprised to see such a high speckled background background and lack of clearly delineated kinetochores in a mock-depleted cell stained for CenpE.

We added the following text in the caption of Fig. S4 to explain the spindle orientation and phase, in particular that in panel c the spindles are vertically oriented, i.e., there was no failure in spindle assembly:

“All spindles were horizontally oriented and in metaphase, except in (c), where the spindles were vertically oriented and in early prometaphase (because Spindly is localized on the kinetochores only during this stage, and this stage is most easily identified in vertically oriented spindles).”

We agree that the images of CENP-E staining in the original manuscript were inadequate, thus we repeated the experiment with another antibody, which resulted in much better staining. The new images and quantification are shown in Supplementary Fig. 4d.

Line 197. This formulation is an example of the tendency to formulate conclusions quite strongly. The Aurora inhibition results reveal a significant decrease in PRC1 and tubulin density, within microtubule bundles. This observation does not prove that “Aurora B activity regulates the number of microtubules within the bridging fibers in metaphase”. Not every bundle is a bridging fiber and disintegration of the bundles in Aurora B inhibited cells is evidence of specific regulation of microtubule numbers.

We toned down this conclusion: “These results suggest that Aurora B activity regulates the number of microtubules within the PRC1-labeled bridging fibers in metaphase.”

Fig.2 F and G. The measurements are presented (and later interpreted) as “mean tubulin intensity in the middle of the bridging fiber in control and Aurora B-inhibited cells”. Yet, no bridging fiber is apparent in the bottom image of panel F. The numbers that are interpreted as a decrease of MT numbers within bridging fibers but the image suggests that a lower number of centromeres associate with bridging fibers.

We agree with the reviewer. We analyzed a larger number of bridging fibers and we also estimated the number of centromeres associated with bridging fibers, as suggested. The revised text reads:

“To confirm this, we measured the tubulin signal intensity in the central part of individual bridging fibers in the region between sister kinetochores in metaphase and found a 56.3% decrease upon Aurora B inhibition, on average (Fig. 2f,g and Supplementary Fig. 3d). In 13.5% (7 out of 52) of these regions between sister kinetochores a bridging fiber was undetectable, i.e., its signal was below the background, and this fraction was larger than in untreated cells, where bridging fibers were undetectable in 1.3% cases (1 out of 78, Fig. 2g). These results suggest that Aurora B activity regulates the number of microtubules within the PRC1-labeled bridging fibers in metaphase.”

Fig.2 I. Approximately 1.5 fold increase in the overall PRC1 density measured via line-scans in cells with displaced Aurora B is puzzling as the bundles are dimmer in Aurora B-inhibited cells. This should be discussed.

We added the following to the corresponding figure caption: “The overall higher anti-PRC1 signal after transfection with CENP-B-INCENP-GFP in comparison with control (no plasmid) is likely due to overexpression of INCENP.”

Fig.3B, C. The same problem as in Fig.2 F,G – there is no bridging fiber associated with this centromere. The image depicts two K-fibers in the classic amphitelic arrangement.

We replaced these images with another, more typical example, and extended the analysis as for Fig. 2f,g:

“CENP-E depletion by siRNA resulted in a 3.1-fold lower intensity of PRC1 in individual bundles in metaphase (n=730 bundles from 14 cells, Fig. 2d,e and Supplementary Fig. 3b,c; see Supplementary Fig. 4 for depletion efficiency) and a 59.3% decrease in the tubulin signal intensity in the central part of individual bridging fibers (Fig. 3b,c and Supplementary Fig. 3d), where in 1.4% cases bridging fibers were undetectable (1 out of 70, Fig. 3c).”

Minor stylistic points

Figure panels are not referenced sequentially. Specifically, the text progresses from Fig.2C to Fig.3, then to the supplementary Fig.3 and back to Fig.2.

The text progresses through Fig. 2 panels sequentially, then to Fig. 3, and then back to Fig. 2d,e. We agree that this is not optimal, but we prefer to keep it this way because it is best to have the PRC1 signal intensity values for all treatments in a single graph (Fig. 2d) and the corresponding images in a single panel (Fig. 2e), so that different treatment can be easily compared with each other and with untreated cells. This is the reason for referring to Fig. 2d,e again after Fig. 3.

‘Squassh’ is spelled as ‘squash’ throughout the manuscript.

This is corrected.

Lines 49-53: There is no proof that effects of Eg5 inhibition are directly linked to the function of this protein within microtubule bundles. Eg5 acts on antiparallel individual microtubules as well. Indeed, references 13-16 do not implicate overlap bundles in the Eg5 phenotype.

We corrected this by deleting “within the bundle” in the context of Eg5 and writing simply that Eg5 acts on antiparallel microtubules: “... inhibition of the Eg5/kinesin-5 motor protein, which slides the antiparallel microtubules apart, leads to monopolar spindles¹³⁻¹⁶.”

Fig.1A. The images depict cells in early prometaphase, late prometaphase, and metaphase. They should be labeled as such. Congression is a process not a stage of mitosis. Further, judging by the width of chromosome spread in axial direction, similar numbers of chromosomes are ‘congressed’ (reside near the spindle equator) in the depicted cells.

The labels are changed as suggested.

Line 179. This statement references two publication that reached somewhat different conclusions. Here is a quote from ref.54 regarding ref.53: “Thus, the association of non-KMT

bundles solely with one pair of sister K-fibers, which has been described in the light microscope (Polak et al., 2017), is not confirmed in ET reconstructions...”. Not a big deal but looks somewhat awkward. Indeed, a more straightforward segue to focusing the chapter on the interplay of kinetochores and PRC1 bundles would be a reference to the recent work by Renda and co-authors (2022) where interactions between kinetochores and bundles during early prometaphase were directly observed.

We agree and have changed the opening sentence of this section as follows (ref. 12 is Renda et al.): “Overlap microtubule bundles crosslinked by PRC1 are found adjacent to kinetochores during the formation of kinetochore fibers in prometaphase¹², as well as in metaphase⁵³.”

REVIEWER 2:

Reviewer #2: In this study, Matkovic et al. study the formation of overlap bundles of microtubules, comprised of antiparallel microtubules extending from opposite spindle halves, within mitotic spindles in human tissue cultured cells. By using STED super-resolution microscopy, the authors show that a transition from a diffuse array of microtubules to a bundled organization occurs between early prometaphase and metaphase. This progressive appearance of bundles is concomitant to chromosome alignment, and is accompanied by increased spacing between bundles and their decoration by the microtubule cross-linker PRC1. Functional analysis revealed that the initial array of microtubules is largely nucleated by the augmin complex. Individual microtubules are then initially bundled by laterally interacting with CENP-E at kinetochores, under the control of the AuroraB kinase, before being cross-linked by PRC1. Importantly, microtubule bundling is independent of Ndc80, and thus of kinetochore fiber formation. By depleting various microtubule motors or kinetochore components, the authors show that microtubule bundles are progressively separated from each other by the steric interactions induced by chromosome alignment on the metaphase plate. They also provide a theoretical model in support of their in experimental findings. Finally, to test the functional relevance of overlap bundles for chromosome segregation, the authors reduced their assembly (by about 50%) by depleting PRC1, and analyzed error correction in cells washed-out of monastrol. This revealed a mild but significant increase in the fraction of lagging kinetochores in cells with decreased overlap bundles (from 2% to 11%). Overall, this is an interesting study, which highlights the mechanism of assembly of an understudied population of microtubule bundles in mitotic spindles, and also provides a potential function for them during chromosome alignment.

Listed below are suggestions and relatively minor points that would improve the final version of this manuscript:

1-Since the authors propose that CENP-E activity is regulated by AuroraB for initial microtubule bundling, it would be interesting to include analysis of the non-phosphorylatable T422A CENP-E mutant (from Kim, Holland et al., Cell, 2010).

This is a great idea. We performed the suggested experiment and included the results in Fig. 3h,i, and in the main text: “Our hypothesis that CENP-E activity, regulated by Aurora B phosphorylation, promotes microtubule bundling predicts that the non-phosphorylatable T422A CENP-E mutant⁶⁵ should lead to decreased bundling. Indeed, replacing endogenous CENP-E with the T422A mutant resulted in a 34% smaller amount of PRC1 on the metaphase spindle, indicating reduced microtubule bundling (n=66 cells from 5 independent experiments, Fig. 3h,i).”

2-To further confirm that overlap bundles can serve as tracks for the CPC to phosphorylate kinetochore targets, an analysis of other phospho-targets (Knl1, Dsn1, Ndc80 or else) could be included in a revised version of this manuscript.

We absolutely agree with this suggestion. However, these data are not crucial for the conclusions of this paper and are more suitable for a separate study focused on the role of PRC1 in error correction.

3-Include Extended Fig. 3H and 3J in Fig. 2 and 3 respectively.

This is now included.

4-Add a control panel in Fig. 3B.

Control panel in Fig. 3B is added.

5-Add images of siKif18A in Fig. 4.

An image is now added.

6-Line 453: edit sentence.

This is corrected.

7-Line 518: edit 'thought'.

This is corrected.

8-Line 573: edit 'combing'.

This is corrected.

REVIEWER 3:

Reviewer #3: This contribution hypothesizes a mechanism for mitotic spindle formation through formation of microtubule bundles mediated by competing attraction/repulsion mechanisms involving kinetochores and chromosomes.

An experimental part using various staining and chromosome modification techniques (e.g., MUG, Ndc80 depletion) attempts to support the hypothesis. Ascribing observations to specific modifications of the system is difficult in the absence of a model of the biomechanics.

The authors introduce a simple one-dimensional model purporting to validate the spindle formation hypothesis. Within the paper itself, the model is perfunctorily described, and later in the Discussion section stated to "have shown" (l. 483) the hypothesis. While fully appreciating the difficulty of constructing mathematical models of the biomechanics that are both insightful and sufficiently simple, a haphazard and careless presentation of the mathematical model does not lend credence to the overall conclusions within this contribution.

In supplementary Theoretical model section, some approaches are questionable, the style of presentation tends to obfuscation, and errors are present in the derivation:

(1) A one-dimensional model along the circumference of the spindle midplane is introduced (Fig. 5), and a partial differential equation is introduced expressing microtubule conservation using a common time dependent Ginzburg-Landau model:

$$\rho_t = M \operatorname{div}(\operatorname{grad}(F))$$

using a hypothesized free energy model containing ρ^4 , ρ^2 , and $(\rho_x)^2$ terms. The choice of a simple proportionality by the mobility M should be discussed, since there are various ways to introduce time-dependence in a Ginzburg-Landau model. The two-dimensional model is reduced to one dimension along the circumferential variable x , valid if the region of interest is thin by comparison to the mean radius. Fig. 4B suggests otherwise: the spindle mid-plane ring is not of negligible thickness, hence some discussion of the two-dimensional model is warranted.

We thank the referee for pointing out the potential ambiguity that may arise from the choice of mobility M_ρ as there are different ways to write down the time evolution of a field within the Landau-Ginzburg model. To clarify the point further we have incorporated the discussion on the choice of a conservative dynamics where the mobility is a natural choice to introduce time dependence in the dynamical equations in the Theoretical Model section:

“As the density conservation is in place, we proceed to write a conserved dynamics of the field ρ ,

$$\frac{\partial \rho}{\partial t} = \frac{-\partial J}{\partial x} \quad (2)$$

where the current is given by $J = -M_\rho \partial \mu / \partial x$. The chemical potential is obtained by minimizing free energy, $\mu = \frac{\delta}{\delta \rho} (\beta F)$. A constant M_ρ denotes mobility of microtubules and in general it is related directly to the diffusivity of the microtubules. Thus, considering the equilibrium properties of the system we obtain the time evolution as,

$$\frac{\partial \rho}{\partial t} = M_\rho \left\{ w \frac{d^2 \rho^3}{dx^2} - \kappa \frac{d^4 \rho}{dx^4} - 2\zeta \psi \frac{d^2 \rho}{dx^2} - 2\alpha \frac{d^2}{dx^2} \sum_{i=0}^{N-1} \rho \delta(x - (2i - 1)d) \right\} \quad (3)$$

Moreover, rapid polymerization and depolymerization of microtubule filaments causes the processes of creation and annihilation of those filaments in the mid plane of the vertically oriented spindle. We assume that microtubules are created with a constant rate ω_{on} and get disassembled with a rate ω_{off} . Finally, we calculate how microtubule density changes in time t by incorporating this nonequilibrium turnover dynamics of microtubules along with the equilibrium processes as,

$$\frac{\partial \rho}{\partial t} = M_\rho \left\{ w \frac{d^2 \rho^3}{dx^2} - \kappa \frac{d^4 \rho}{dx^4} - 2\zeta \psi \frac{d^2 \rho}{dx^2} - 2\alpha \frac{d^2}{dx^2} \sum_{i=0}^{N-1} \rho \delta(x - (2i - 1)d) \right\} + \bar{\omega}_{on} - \omega_{off} \rho \quad (4)$$

Nucleation rate per unit length is given by $\bar{\omega}_{on} = \frac{\omega_{on}}{L}$.

Also, in our revised manuscript, we have addressed the important concern raised by the referee regarding the two dimensional extension of our one dimensional model:

“To describe the dynamical transition from array like distribution of microtubules to well organized multiple bundles during prometaphase, we consider both equilibrium and non-equilibrium processes involving microtubules, crosslinking proteins, chromosomes, and kinetochores. The model we construct is placed in an one dimensional geometry with spatial coordinate x , taken from the part of the circumference of the disc, L , that is obtained by taking a vertical cross section at the mid plane of a prometaphase spindle as shown in Fig. 5a. We choose a one dimensional geometry as it substantially reduces the complexity of the problem. One dimensional geometry preserves the major features of the system that we study because in early prometaphase the kinetochores are predominately distributed along a line that forms a circle on the periphery of the prometaphase spindle (see Fig. 2b).”

(2) The model contains five independent parameters w , κ , ζ , α , ω . Non-dimensional values are given in the discussion of Fig. 5 with reference to the Methods. No information is available within the Methods section on why these particular parameters were chosen. Absent some link between the chosen model parameters and the biophysics of the system, only qualitative behavior conclusions can be drawn from the theoretical model.

We agree with the referee that the phenomenological parameters are not connected with biological processes and thus cannot be linked to the quantitative behavior of the system. In our revised manuscript we use 6 physical parameters with dimension. We estimate 4 parameters from experimental data available in literature, whereas the other 2 we use as adjustable parameters, as described in Methods:

“Connections between Model Parameters and Biological Processes. The parameters ω_{on} and ω_{off} are related to microtubule polymerization and depolymerization respectively, across the circumferential ring of the midplane cross section. To calculate ω_{on} we first estimate the current of plus-ends of growing microtubules, $J_c = \rho_c v_g$, from the linear density of the microtubule in the midzone, ρ_c , and the microtubule growth velocity, v_g . Previous measurements of the EB1 comets that are passing through the spindle midplane of thickness of $0.5 \mu m$ reveal that around 27000 spots were present in the zone during 75 frames in 3 cells, yielding the $\rho_c = 27000/(0.5 \times 75 \times 3) \mu m^{-1}$ (Fig. 9b from Ref.⁹⁷). From the same measurements, the growth velocity was $v_g = 0.465 \mu m s^{-1}$ (Fig. 6a from Ref.⁹⁷), yielding $J_c = 110 s^{-1}$. The circumferential ring of the midplane cross section has radius $5 \mu m$, but the area where kinetochores lay is $1 \mu m$ thick and it represents approximately 1/3 of the midplane cross sectional area, giving rise $\omega_{on} = J_c/3 = 40 s^{-1}$. The other parameter, ω_{off} , is estimated from the total number of microtubules in the spindle, $N = J_c/\omega_{off} = 6300$, based on the electron tomography data³⁸, giving $\omega_{off} = 0.02 s^{-1}$.

The parameter ζ is related to the binding energy of one PRC1 molecule that crosslinks a pair of microtubules. We estimate binding energy of a single PRC1 as a probability of two microtubules to get close enough so that a single PRC1 can crosslink them as $(\rho x_{PRC1})^2$, which is multiplied by a binding energy of that microtubule pair formation, ϵ . Thus, the parameter related to the binding energy has a value $\zeta = \beta \epsilon x_{PRC1}^2$, where x_{PRC1} denotes the length of a PRC1 dimer. We estimate the binding energy ϵ to the order of $10 k_B T$ as it should be strong enough to resist thermal fluctuations and at the same time must be weak enough so that PRC1 can undergo spontaneous dissociation as seen experimentally²⁹. The unstretched length of the PRC1 dimer $x_{PRC1} = 40 nm$, based on Ref.⁹⁸, yielding $\zeta = 0.016 \mu m^2$.

The kinetochore-microtubule coupling strength, α , is associated with the binding energy of the microtubules to the kinetochore. We estimate it as the coupling strength between a single microtubule and kinetochore, $\bar{\epsilon}$, multiplied by the number of available binding, n_{site} , and normalized by the saturating microtubule surface density at the binding sites, ρ_{site} , yielding $\alpha = n_{site} \beta \bar{\epsilon} / \rho_{site}$. We estimate the binding energy per microtubule to be $\bar{\epsilon} = 10 k_B T$, which is similar to the binding energy of a single motor protein to the microtubule⁹⁹, the number of available binding sites $n_{site} = 10$ is estimated from the number of microtubules bound to one kinetochore³⁸, and the saturating microtubule surface density is $\rho_{site} = 80 \mu m^{-2}$ (Ref. ³⁸), yielding $\alpha = 1.25 \mu m^2$.”

(3) The free energy model itself contains fixed kinetochore positions (the Dirac delta terms). Since these positions are fixed, it is not surprising that model parameters can be chosen to show "bundling" or, in mathematical terms, attraction points - they were built into the model

from the start. This qualitative behavior should be completed by discussion of the model parameter space, and of the case in which kinetochore positions themselves change in time (as suggested by, e.g., Fig. 3, F).

To address this question, we introduce effective free energy and explore the stability of the system by comparing the effective free energy of the two systems. In one of them, kinetochores are equally spaced, whereas in other inter kinetochore distance is nonuniform. Our calculations reveal that the homogeneously distributed kinetochores are more stable. Therefore, the whole system would dynamically tend to go towards evenly spaced kinetochore distributions. We added a sentence in the main text: “Kinetochores are equidistantly spaced, and we find that for this choice of kinetochore distributions our system is in a stable equilibrium.” and the following text in Methods:

“Effective free energy of the system. Once the microtubule density profile ρ_f is obtained, we construct an effective free energy description by putting back ρ_f into Eq.(1). As the time-dependent Ginzburg-Landau equation contains nonequilibrium contribution of microtubule turnover that also governs the shape of the density profile, we use the term effective free energy, instead of free energy. We use this approximate free energy description of the microtubule-crosslinker-kinetochore system to calculate the stability of a uniform lattice spacing.

To check the stability of the system, we compare the effective free energy of a lattice with a uniform lattice spacing and that with a nonuniform spacing in the system consisting of two kinetochores. The lattice stretches from $-2d$ to $+2d$ with total length $L = 4d$ and a periodic boundary condition is imposed on it. In the uniformly spaced lattice, kinetochores are at positions $\pm d$, yielding the distance between kinetochores $2d$. In the deformed lattice kinetochores are placed at positions $-d + \epsilon$ and $d - \epsilon$, with two different distances between kinetochores, $2d - 2\epsilon$ and $2d + 2\epsilon$, where ϵ denotes small displacement from the uniform lattice. Thus, the effective free energy of the deformed lattice is given by,

$$\beta F_{eff} = \int_{-2d}^{2d} dx \left\{ \frac{w}{4} \rho_f^4 + \frac{\kappa}{2} \left(\frac{\partial \rho_f}{\partial x} \right)^2 - \zeta \rho_f \psi \rho_f - \alpha \rho_f^2 [\delta(x + d - \epsilon) + \delta(x - d + \epsilon)] \right\}, \quad (12)$$

We evaluate this integral by splitting the integration interval onto three subintervals: $[-2d, -d + \epsilon]$, $[-d + \epsilon, d - \epsilon]$, and $[d - \epsilon, 2d]$. Approximate solutions in these intervals can be written as $\rho_f = \rho_0 \pm \Delta\rho$, where ρ_0 denotes microtubule density for the uniformly spaced lattice and $\Delta\rho$ denotes a small distortion from microtubule density of the uniformly spaced lattice. The plus sign corresponds to the first and the third intervals, and the minus sign corresponds to the second interval.

The lowest correction term is of the order of ϵ^2 . After evaluating the Eq. (12), we find that the only nontrivial contribution comes from terms that represent kinetochore-microtubule attraction, δ functions, yielding the difference in the effective free energy $\beta \Delta F_{eff} = [2\alpha \rho_0 (d^2 \rho_0 / dx^2)]_{x=d} \epsilon^2$. For the parameters explored in Fig. 5, the microtubule density ρ_0 and its second derivative have positive values. Thus, the effective free energy difference has a

positive value for a small displacement of kinetochores from the uniform lattice. This result indicates that the evenly spaced kinetochore lattice is placed at the local minimum of effective free energy and that for this choice our system is in stable equilibrium.”

(4) The model itself contains annoying if minor errors:

(a) kinetochores are equally-spaced along the circumference, but the free energy contains only a contribution from that of indices $\pm(2n+1)$. Are only nearest neighbors considered, or is a sum missing?

(b) There is an extra M_ρ factor term in the third right-hand term of the unnumbered equation above equation (2)

(c) The non-dimensionalization is incorrect. Using the stated reference quantities the coefficient of the first right-hand term of (2) becomes

$\omega_{\text{off}} M \tilde{M}^{4/3}$

instead of $\tilde{M} M$ as in the unnumbered equation above.

Regarding the point (a), we have corrected the typo by adding summation over index i , and the free energy in Eq. (1) now reads:

$$\beta F[\rho] = \oint_{\text{entire } L} dx \left\{ \frac{w}{4} \rho^4 + \frac{\kappa}{2} \left(\frac{\partial \rho}{\partial x} \right)^2 - \zeta \rho \psi \rho - \sum_{i=0}^{N-1} \alpha \rho^2 \delta[x - (2i - 1)d] \right\}, \quad (1)$$

This typo appeared earlier in this equation, but not in our calculations and therefore this correction did not affect our results and conclusions.

Regarding the point (b), we have corrected the typo in the manuscript by removing an extra factor M_ρ .

Regarding the (c), we have rewritten our model by using parameters and variables with dimensions, so in the revised manuscript such typos related to non-dimensionalization do not appear.

(5) The non-linear time-dependent Landau-Ginzburg model is subsequently linearized and treated in an unstated nearest-neighbor hypothesis. The elementary and tedious discussion of the solution of a linear fourth-order ODE can be streamlined, and more effort dedicated to determining the range of validity of the linearization. If this were a small perturbation around the bundled state, the model might be valid. But the whole point of the model is to support the hypothesis of evolution from a uniform microtubule distribution to one concentrated at the fixed kinetochore locations. Nonlinear dynamical systems can exhibit complicated phase space trajectories in this case.

Following the suggestion by the reviewer, we have replaced the approximated linearized calculation with a power series solution without linearization of the differential equation. This

approach is also approximate because we calculated up to the 6th order term of the power series. We tested the validity of this approach by comparing the result from 4th order with 6th order and found that the difference is around 5% for $d=2 \mu\text{m}$ and $\alpha = 10 \mu\text{m}^2$, suggesting that numerical error is below 5%. This error reduces to 0.5% when the inter kinetochore distance is $1 \mu\text{m}$, keeping all other parameters fixed. We have incorporated the following section in Methods:

“Numerical solution of the steady state nonlinear differential equation by power series expansion. The steady state density ρ_f is obtained by setting $\frac{\partial \rho_f}{\partial t} = 0$ in Eq. (4) as,

$$M_\rho \left\{ w \frac{d^2 \rho_f^3}{dx^2} - \kappa \frac{d^4 \rho_f}{dx^4} - 2\zeta\psi \frac{d^2 \rho_f}{dx^2} - 2\alpha \frac{d^2}{dx^2} \sum_{i=0}^{N-1} \rho_f \delta(x - (2i-1)d) \right\} + \bar{\omega}_{on} - \omega_{off} \rho_f = 0 \quad (5)$$

This is an ordinary nonlinear differential equation with the highest order 4 and with cubic nonlinearity. To solve it, we use a power series expansion given by

$$\rho_f(x) = \sum_{j=0}^{\infty} \frac{a_j}{j!} x^j \quad (6)$$

where a_j are the coefficients of the expansion. Because of the nonlinearity of Eq. (5), the expression of higher order terms becomes more complex as it couples larger number different coefficients in the expansion. Thus, we search for an approximative solution which can be calculated by several lowest orders of the expansion.

The system consists of periodically placed attractive δ functions which has a mirror symmetry with respect to $x = 0$. Thus, we search for solutions that have a mirror symmetry, $\rho_f(-x) = \rho_f(x)$. This condition implies that in Eq. (6) only even powers in the expansion will survive. To find an approximative solution, we truncate Eq. (6) until 6th order and the expansions are given by

$$\rho_f(x) = a_0 + \frac{a_2}{2!} x^2 + \frac{a_4}{4!} x^4 + \frac{a_6}{6!} x^6 \quad (7)$$

By inserting this expansion into Eq. (5) we obtain a set of recursive relations between coefficients that read as,

$$a_4 = \frac{1}{\kappa M_\rho} (3wM_\rho a_0^2 a_2 - 2\zeta\psi M_\rho a_2 + \bar{\omega}_{on} - \omega_{off} a_0) \quad (8)$$

$$a_6 = \frac{2}{\kappa M_\rho} \left[wM_\rho \left(9a_0 a_2^2 + \frac{3}{2} a_0^2 a_4 \right) - \zeta\psi M_\rho a_4 - \frac{\omega_{off}}{2} a_2 \right] \quad (9)$$

Proceeding in the same way one can obtain higher order corrections into the microtubule density profile ρ_f .

To fix the two remaining integration constants, a_0 and a_2 , we use the boundary conditions. The boundary condition that involves the Dirac delta function, which reflects as a discontinuity in the first derivative of ρ_f at $\pm d$,

$$\kappa \left[\frac{d\rho_f}{dx} \Big|_{x=-d+\epsilon} - \frac{d\rho_f}{dx} \Big|_{x=d-\epsilon} \right] = -2\alpha\rho_f(d), \quad (10)$$

in the limit $\epsilon \rightarrow 0$. The other boundary condition takes into account conservation of microtubule numbers, which reads

$$\int_{-d}^d \rho_f(x) dx = \frac{2d\bar{\omega}_{on}}{\omega_{off}}, \quad (11)$$

By inserting expansion given by Eq. (7) and expression from Eqs. (10) and (11), we fix the two remaining integration constants, a_0 and a_2 .

We tested the validity of this approach by comparing the result from 4th order with 6th order and found that the difference is around 5% for $d=2 \mu m$ and $\alpha = 10 \mu m^2$, suggesting that numerical error is below 5%. This error reduces to 0.5% when the inter-kinetochore distance is $1 \mu m$, keeping all other parameters fixed.”

REVIEWER COMMENTS

Reviewer #1 (Remarks to the Author):

The authors have convincingly addressed all but two of this reviewer's concerns. The remaining issues are the question of whether the live recordings (Fig.1, S1, Video 1) were obtained under photodamaging conditions, and the quality of immunostaining presented in Fig.S4.

The model constructed in the manuscript is based on the dynamic reorganization of PRC1 from the relatively diffused pattern at NEB to discrete bundles in prometaphase. Essential for justification of the model is whether the observed PRC1 behavior occurs during normal, i.e., fully functional spindle assembly. Noteworthy is that photodamage is known to result in excessive bundling of spindle microtubules so adverse effects of excessive illumination on the behavior of PRC1 bundles are likely. A proper way to address this concern is to demonstrate that cells subjected to imaging conditions necessary for high-resolution recordings subsequently progress into anaphase. Technically, this can be easily achieved as after 10 minutes of recording at 5-s temporal resolution, intervals between frames can be increased to something like a minute. If the cell initiates anaphase in ~30 min, the recording is physiologically safe and the PRC1 dynamics are likely normal. When I raised this issue in my original review, I expected that a proof of normal progression through mitosis will be provided. A model for PRC1 behavior under conditions that do not lead to a normal functional spindle is not that interesting.

The second remaining concern is the clearly abnormal Spindly staining pattern in Fig.S4c. Distribution of this protein in HeLa cells is well established and it differs significantly from the one presented in the manuscript (e.g., compare with Fig.1A in the original characterization of Spindly in HeLa cells – Gassmann et al., 2010). In the rebuttal letter and in the revised legend for Fig.S4, the authors state that Fig.S4c illustrates a vertically oriented spindle during early prometaphase cell. However, this explanation contradicts their own data. In the manuscript, early metaphase is defined as the stage when “chromosomes are arranged like flower petals along the edge of the nascent spindle^{49,50}, [and] the majority of spindle body microtubules appeared as a diffuse array with a few bundles present mainly at the edge” (lines 96-99). In contrast to this description (as well as to Fig.1), in Fig. S4c kinetochores are intermixed with numerous PRC1 bundles – this is not an early prometaphase cell and the magenta spots are not Spindly-positive kinetochores.

One more purely technical issue that the authors might want to clarify:

Lines 770-772 There appears to be a couple of technical problems here. First, the formulation of the ‘microtubule-stabilizing buffer’ referenced to Heuser and Kirschner (1980) is peculiar. Indeed, the buffer described by the authors cannot stabilize microtubules because 100 mM Mg⁺⁺ would be completely chelated by 100 mM EDTA within the buffer. Heuser and Kirchner used Mg⁺⁺ in fivefold excess to EDTA and supplemented their buffer with 4-M glycerol to preserve microtubules. Conventional mt-stabilizing buffers (PEM/BRB80) combines Mg (to stabilize mts) and EGTA. Second, there is no Heuser and Kirschner, 1980, article in the list of references. I guess that the reference was to the platinum replicas paper in the JCB.

Reviewer #2 (Remarks to the Author):

All my comments or suggestions have been satisfactorily addressed. I therefore recommend publication of the revised manuscript.

Reviewer #3 (Remarks to the Author):

Comments on Theoretical Model

The revised manuscript maintains the hypothesis of a linear microtubule (MT) density flux dependence on the gradient $J = -M_{\rho} \rho_x$, through a constant mobility M_{ρ} . The effective free energy is a cogent formulation of the possible factors driving MT density changes.

Since the system contains both attractive and repulsive terms, steady-states with local attractors are possible, and authors present reasonable parameter values for which such local MT density increases are obtained. Though the parameters themselves are affected by the uncertainty prevalent in estimates of physical interactions in biological systems, the theoretical model is indeed supportive of the main hypothesis of the paper.

No errors were found in the derivations or calculations.

The only remark would be for authors to state observations/conclusions more cautiously:

I. 396 states, "a minimal model is an optimal tool". Not really. A model that would incorporate possible two-dimensional perturbations of the configuration would be more informative. Granted, the effort to derive/analyze such a model would be significant. Perhaps a better formulation is 'a minimal model is a useful tool'.

I. 403 states the system "is in a stable equilibrium". This is an overly broad statement. For the chosen parameter values, and the restriction to 1D perturbations, the system was found to be stable. Perhaps a more cautious statement is on the lines of 'stable equilibria are found for reasonable parameter choices'.

I. 79-81 states "Our work thus reveals that kinetochores and chromosomes together with crosslinkers drive coarsening of an initially uniform microtubule array into neatly organized overlap bundles". Again, this is too strong a statement for the given model. Dynamics of the system were not considered. The theoretical model simply contains an analysis of the steady-state. When the system is restricted to a single field variable, i.e., the MT density, it can be expected that it will evolve to the steady state from generic initial conditions. However, it is unlikely that the true dynamics are represented just by MT density. Both kinetochore density and cross-linker density are additional field variables (possibly also, chromosome density). The resulting mathematical description would be a nonlinear dynamical system, and whether it would evolve to the equilibria considered in this contribution is an open question.

I. 509: "By combining STED microscopy, a live-cell bundling assay and theory, we have shown that spindle assembly occurs through a transition in which the initially uniform microtubule array is remodeled into bundles (Fig. 7a-c)."

As above, given the model limitations, "have shown" is too broad a statement.

REVIEWER COMMENTS

Reviewer #1 (Remarks to the Author)

The authors have convincingly addressed all but two of this reviewer's concerns. The remaining issues are the question of whether the live recordings (Fig.1, S1, Video 1) were obtained under photodamaging conditions, and the quality of immunostaining presented in Fig.S4.

The model constructed in the manuscript is based on the dynamic reorganization of PRC1 from the relatively diffused pattern at NEB to discrete bundles in prometaphase. Essential for justification of the model is whether the observed PRC1 behavior occurs during normal, i.e., fully functional spindle assembly. Noteworthy is that photodamage is known to result in excessive bundling of spindle microtubules so adverse effects of excessive illumination on the behavior of PRC1 bundles are likely. A proper way to address this concern is to demonstrate that cells subjected to imaging conditions necessary for high-resolution recordings subsequently progress into anaphase. Technically, this can be easily achieved as after 10 minutes of recording at 5-s temporal resolution, intervals between frames can be increased to something like a minute. If the cell initiates anaphase in ~30 min, the recording is physiologically safe and the PRC1 dynamics are likely normal. When I raised this issue in my original review, I expected that a proof of normal progression through mitosis will be provided. A model for PRC1 behavior under conditions that do not lead to a normal functional spindle is not that interesting.

Authors: To assess whether cells subjected to imaging conditions necessary for high-resolution recordings subsequently progress into anaphase, we performed new experiments in which we imaged the cells as in the “bundling assay” at 5.4 s intervals, and subsequently at 5 min intervals to measure the time of anaphase onset. As control, we imaged the same cell line “gently” at 5 min intervals from early prometaphase to anaphase. We found that the cells imaged for the “bundling assay” entered anaphase at similar times as control cells, indicating normal progression through mitosis (Supplementary Fig. 1g,h).

To further assess whether the observed PRC1 behavior occurs during normal, fully functional spindle assembly, we performed additional experiments in which we used untransfected HeLa cells and immunostained them for PRC1, together with DNA staining to identify the phases of mitosis. These cells showed similar reorganization of PRC1 as the cells imaged by the bundling assay (Supplementary Fig. 1i,j), suggesting that the observed PRC1 dynamics occurs during fully functional spindle assembly.

These new experiments are shown in Fig. S1 and described in the main text on page 4: “The intense imaging protocol did not affect spindle functioning, given that the spindles subjected to the bundling assay entered anaphase at similar times to control spindles that were imaged at 5-minute intervals (Supplementary Fig. 1g,h), and the network-to-bundles transition found in the bundling assay was consistent with STED images of PRC1-GFP (Fig. 1a and Supplementary Fig. 1a) and confocal images of untransfected cells immunostained for PRC1

(Supplementary Fig. 1i,j). The latter result also implies that the dynamics of PRC1-GFP reflects the dynamics of endogenous PRC1. Thus, a network-to-bundles transition of PRC1 occurs during fully functional spindle assembly.”

The second remaining concern is the clearly abnormal Spindly staining pattern in Fig.S4c. Distribution of this protein in HeLa cells is well established and it differs significantly from the one presented in the manuscript (e.g., compare with Fig.1A in the original characterization of Spindly in HeLa cells – Gassmann et al., 2010). In the rebuttal letter and in the revised legend for Fig.S4, the authors state that Fig.S4c illustrates a vertically oriented spindle during early prometaphase cell. However, this explanation contradicts their own data. In the manuscript, early metaphase is defined as the stage when “chromosomes are arranged like flower petals along the edge of the nascent spindle^{49,50}, [and] the majority of spindle body microtubules appeared as a diffuse array with a few bundles present mainly at the edge” (lines 96-99). In contrast to this description (as well as to Fig.1), in Fig. S4c kinetochores are intermixed with numerous PRC1 bundles – this is not an early prometaphase cell and the magenta spots are not Spindly-positive kinetochores.

Authors: The spindles in Fig. S4c were indeed in early prometaphase, but we agree that this was not obvious from the maximum-intensity projections of vertically oriented spindles. To clarify this issue, we now added images of the central plane (labeled “one z”) including DAPI staining of the same spindles from Fig. S4c next to the maximum-intensity projections (labeled “max z”). In the central planes it is evident that chromosomes are arranged like flower petals along the edge of the nascent spindle, indicating that this is an early prometaphase spindle. The Spindly spots are found at the edge of the spindle, suggesting that they mark Spindly-positive kinetochores.

To further demonstrate Spindly depletion, we performed new immunostaining experiments in which we measured Spindly signal in horizontally oriented prometaphase spindles. As expected, we observed Spindly at kinetochores and spindle poles in control cells. The Spindly signal on the spindle was very weak in Spindly siRNA cells, confirming efficient (85%) Spindly depletion. The images and quantification of these new experiments are shown in Fig. S4c and explained in the corresponding figure caption.

One more purely technical issue that the authors might want to clarify:

Lines 770-772 There appears to be a couple of technical problems here. First, the formulation of the ‘microtubule-stabilizing buffer’ referenced to Heuser and Kirschner (1980) is peculiar. Indeed, the buffer described by the authors cannot stabilize microtubules because 100 mM Mg⁺⁺ would be completely chelated by 100 mM EDTA within the buffer. Heuser and Kirchner used Mg⁺⁺ in fivefold excess to EDTA and supplemented their buffer with 4-M glycerol to preserve microtubules. Conventional mt-stabilizing buffers (PEM/BRB80) combines Mg (to stabilize mts) and EGTA. Second, there is no Heuser and Kirschner, 1980, article in the list of references. I guess that the reference was to the platinum replicas paper in the JCB.

Authors: We removed the expression ‘microtubule-stabilizing buffer’ and the reference to Heuser and Kirschner (1980). We added the correct references from which this protocol was taken: Chozinski, T. J. et al. Expansion microscopy with conventional antibodies and fluorescent proteins. *Nat. Methods* 13, 485–488 (2016); and Ponjavić, I., Vukušić, K. & Tolić, I. M. Expansion microscopy of the mitotic spindle. *Methods Cell Biol.* 161, 247–274 (2021).

Reviewer #2 (Remarks to the Author):

All my comments or suggestions have been satisfactorily addressed. I therefore recommend publication of the revised manuscript.

Authors: Thank you.

Reviewer #3 (Remarks to the Author):

Comments on Theoretical Model

The revised manuscript maintains the hypothesis of a linear microtubule (MT) density flux dependence on the gradient $J = -M_{\rho} \rho_x$, through a constant mobility M_{ρ} . The effective free energy is a cogent formulation of the possible factors driving MT density changes.

Since the system contains both attractive and repulsive terms, steady-states with local attractors are possible, and authors present reasonable parameter values for which such local MT density increases are obtained. Though the parameters themselves are affected by the uncertainty prevalent in estimates of physical interactions in biological systems, the theoretical model is indeed supportive of the main hypothesis of the paper.

No errors were found in the derivations or calculations.

Authors: We thank the reviewer for both reports, which helped us to substantially improve the manuscript, and for the overall positive evaluation of the revised version.

The only remark would be for authors to state observations/conclusions more cautiously:

l. 396 states, "a minimal model is an optimal tool". Not really. A model that would incorporate possible two-dimensional perturbations of the configuration would be more informative. Granted, the effort to derive/analyze such a model would be significant. Perhaps a better formulation is 'a minimal model is a useful tool'.

Authors: We agree and have replaced “optimal” with “useful”.

l. 403 states the system "is in a stable equilibrium". This is an overly broad statement. For the chosen parameter values, and the restriction to 1D perturbations, the system was found to be stable. Perhaps a more cautious statement is on the lines of 'stable equilibria are found for reasonable parameter choices'.

Authors: We revised this sentence: "Kinetochores are equidistantly spaced, and we find stable equilibria for this choice of kinetochore distributions and reasonable parameters."

l. 79-81 states "Our work thus reveals that kinetochores and chromosomes together with crosslinkers drive coarsening of an initially uniform microtubule array into neatly organized overlap bundles". Again, this is too strong a statement for the given model. Dynamics of the system were not considered. The theoretical model simply contains an analysis of the steady-state. When the system is restricted to a single field variable, i.e., the MT density, it can be expected that it will evolve to the steady state from generic initial conditions. However, it is unlikely that the true dynamics are represented just by MT density. Both kinetochore density and cross-linker density are additional field variables (possibly also, chromosome density). The resulting mathematical description would be a nonlinear dynamical system, and whether it would evolve to the equilibria considered in this contribution is an open question.

Authors: We revised this sentence to emphasize that the conclusions are mainly based on experiments: "Thus, our experiments, supported by the theoretical model, reveal that kinetochores and chromosomes together with crosslinkers drive coarsening of an initially uniform microtubule array into neatly organized overlap bundles, which not only help spindle assembly but also promote error-free mitosis."

l. 509: "By combining STED microscopy, a live-cell bundling assay and theory, we have shown that spindle assembly occurs through a transition in which the initially uniform microtubule array is remodeled into bundles (Fig. 7a-c)."

As above, given the model limitations, "have shown" is too broad a statement.

Authors: We have removed "theory" from this sentence because this paragraph is related to the experiments, whereas the model is discussed in the next paragraph.

REVIEWER COMMENTS

Reviewer #1 (Remarks to the Author):

The revision appropriately addresses my concerns and I recommend accepting the manuscript for publication.

Reviewer #3 (Remarks to the Author):

All reviewer comments on the theoretical model have been addressed. The paper is a good contribution to understanding the mechanism of spindle formation during mitosis.